# Fate-mapping post-hypoxic tumor cells reveals a ROS-resistant phenotype that promotes metastasis

Inês Godet [1,2], Yu Jung Shin[1], Julia A. Ju[1,2], I Chae Ye[1], Guannan Wang[1] & Daniele M. Gilkes [1,2,3]*

Hypoxia is known to be detrimental in cancer and contributes to its development. In this work, we present an approach to fate-map hypoxic cells in vivo in order to determine their cellular response to physiological $O_2$ gradients as well as to quantify their contribution to metastatic spread. We demonstrate the ability of the system to fate-map hypoxic cells in 2D, and in 3D spheroids and organoids. We identify distinct gene expression patterns in cells that experienced intratumoral hypoxia in vivo compared to cells exposed to hypoxia in vitro. The intratumoral hypoxia gene-signature is a better prognostic indicator for distant metastasis-free survival. Post-hypoxic tumor cells have an ROS-resistant phenotype that provides a survival advantage in the bloodstream and promotes their ability to establish overt metastasis. Post-hypoxic cells retain an increase in the expression of a subset of hypoxia-inducible genes at the metastatic site, suggesting the possibility of a 'hypoxic memory.'

[1] Department of Oncology, The Sidney Kimmel Comprehensive Cancer Center, The Johns Hopkins University School of Medicine, Baltimore, MD 21231, USA. [2] Department of Chemical and Biomolecular Engineering, The Johns Hopkins University, Baltimore, MD 21218, USA. [3] Cellular and Molecular Medicine Program, The Johns Hopkins University School of Medicine, Baltimore, MD 21231, USA. *email: dgilkes1@jhu.edu

Hypoxia is an intensively investigated condition with profound effects on cell metabolism, migration, and angiogenesis during development and disease[1–4]. Pathologies, including cardiovascular disease, ischemia, fibrosis, and cancer, can be severely affected by hypoxia[5,6]. Hypoxia occurs in 90% of solids tumors due to limitations in diffusion and inefficient vasculature and has been identified as an adverse indicator for patient prognosis independent of clinical stage at diagnosis[7,8]. Data describing the pretreatment oxygenation status of solid tumors demonstrated that the mean partial pressure of oxygen ($PO_2$) in breast tumors ranges from 2.5 to 28 mmHg, with a median value of 10 mmHg, as compared with 65 mmHg in normal human breast tissue[9]. $PO_2$ values of <10 mmHg have been associated with an increased risk of metastasis and mortality[10–12].

Oxygen deprivation causes the stabilization of the hypoxia-inducible subunits (HIF-1α and HIF-2α) that regulate the cellular response to hypoxia[13]. HIF-1 or HIF-2 function as heterodimeric proteins composed of the $O_2$-regulated HIF-1α or HIF-2α subunit, and a constitutively expressed HIF-1β subunit[14]. The HIF-1 heterodimer recognizes a 5′-ACGTG-3′ enhancer sequence on DNA to transcriptionally regulate more than a thousand genes[15,16]. The vast majority of studies on hypoxia-regulated gene expression have been conducted in an in vitro setting by exposing cells to 1% $O_2$ for short periods of time.

Both direct and indirect methods have previously been used to identify hypoxic tissue regions in vivo. The direct measurement of $O_2$ in human tumor tissue has been performed by using $O_2$ electrodes[17] or fiber-optic probes[18], but these methods lack the ability to visualize, isolate, or separate individual hypoxic cells. Indirect $O_2$ measurement methods include immune labeling of endogenous markers, such as transcriptional targets of the hypoxia-inducible factors (HIFs) detected with immunohistochemical analysis[19], or by using exogenous 2-nitroimidazole probes, such as pimonidazole, that bind covalently to thiol molecules in hypoxic tissue[20]. While these methods have the ability to identify cells or tissue regions that are hypoxic when the tissue is harvested and fixed, they are limited by the sensitivity and specificity of the biomarker to the hypoxic stimulus, cellular turnover rates, as well as artifacts caused by fixation (Fig. 1a). Standard $O_2$ imaging techniques have been applied clinically, and include X-ray imaging, computed tomography, ultrasound imaging, optical imaging, nuclear imaging, and magnetic resonance imaging[21], which assess tissue voxels but not individual cells. Recently, several groups have utilized HIF-binding sequences to transcriptionally control the expression of fluorescent proteins under hypoxia in order to detect hypoxic cells[22–27]. While these methods are useful to identify cells that are transiently hypoxic in vivo, they cannot be reliably used to track the fate of hypoxic cells that undergo reoxygenation in the bloodstream and lung[28,29]. Therefore, we propose a system to permanently mark cells when they become hypoxic by triggering a fluorescent switch from DsRed to GFP. We demonstrate that our system has a tightly controlled response to low $O_2$ levels (<0.5%), which are comparable to $O_2$ levels measured in human tumors. We characterize the system in 2D and 3D, as well as in orthotopic and mouse models of breast cancer. In parallel studies, we generate a transgenic mouse whose cells permanently switch from tdTomato to GFP expression when they experience hypoxia. We then create a triple-transgenic mouse that develops spontaneous breast cancer to fate-map hypoxic cells during tumor progression. We further show that in vitro studies do not recapitulate the response to intratumoral hypoxia in vivo. Gene expression signatures developed from cells that experience intratumoral hypoxia in vivo serve as better prognostic indicators than those established by using in vitro hypoxic exposure. By mechanistically evaluating the potential of post-hypoxic tumor cells to accomplish each individual step in the metastatic cascade separately, we demonstrate that hypoxia-exposed cells have an enhanced ability to invade/intravasate and to survive in the bloodstream, but they are not more efficient at extravasating or proliferating at the metastatic site compared with their oxygenated counterparts. We further demonstrate that exposure to chronic hypoxia in vivo endows cancer cells with a ROS-resistant phenotype that promotes their survival upon reoxygenation in the bloodstream, increasing the probability of successful metastasis. Targeting this ROS-resistant phenotype may provide a therapeutic strategy to successfully prevent metastasis.

## Results

**Designing and characterizing a hypoxia fate-mapping system**. To detect hypoxia in primary tumor tissue, we used Hypoxyprobe™ (Fig. 1a) to detect hypoxic tumor regions that contain <1% $O_2$. Hypoxic regions were localized adjacent to necrotic tissue, far from blood vessels that diffuse $O_2$. Although hypoxia has been linked to worse outcomes for cancer patients, the localization of hypoxic cells adjacent to regions of necrosis may suggest that given sufficient time, hypoxic cells will become necrotic. To reconcile these seemingly contrary observations, we generated a method to permanently mark hypoxic cells and follow their fate during tumorigenesis. By using a lentiviral delivery approach, we generated a dual-vector hypoxia fate-mapping system. Vector 1 contains an altered Cre gene modified by the addition of an oxygen-dependent degradation domain (ODD) under the transcriptional control of a synthetic HIF–DNA binding sequence (HRE) (Fig. 1b). Vector 2 expresses a red fluorescent reporter protein (DsRed) with a stop codon flanked by tandem loxP sites preceding a gene encoding a green fluorescent protein (GFP). Under hypoxia, HIF stabilization causes the transcriptional activation of genetically modified Cre, leading to the cleavage of DsRed and permanent GFP expression. The number of repetitive HRE sequences was experimentally determined to ensure an irreversible DsRed to GFP reporter switch in cells exposed to no >0.5% $O_2$ (the median $O_2$ concentration that has been measured in human breast tumors), but did not change under physiological conditions that mimic normal breast tissue (8% $O_2$)[9] (Supplementary Fig. 1a). MDA-MB-231, MCF7, and 4T1 breast cancer cells were stably transduced with the lentiviral vector system (Fig. 1b) and displayed similar results (Fig. 1c–e and Supplementary Fig. 1b, c). The timing and sensitivity to variable $O_2$ concentrations was characterized by live-cell time-lapse imaging, flow cytometry, and immunoblot analysis, and was consistent among MDA-MB-231 (Fig. 1f, Supplementary Fig. 1b, d, e and Supplementary Movie 1), MCF7, and 4T1 cell lines (Supplementary Fig. 1f–j).

In parallel studies, we developed a transgenic mouse that similarly expresses CRE in a hypoxia-dependent manner (vector 1) (Figs. 1b and 2a, b). Recently, a transgenic mouse expressing a constitutive reporter that drives a Cre gene containing an oxygen-dependent domain inducible by tamoxifen administration was reported[30,31]. We designed our mouse model to have $O_2$ control at the transcriptional and translational levels, which prevented leakiness of the Cre gene and negated the need for tamoxifen control. By using targeted locus amplification (TLA) and next-generation sequencing[32], we identified the exact integration site, copy number, sequence, and orientation of the transgene (Fig. 2b and Supplementary Fig. 2a, b). We bred our hypoxia-dependent Cre-expressing mouse to a tdTomato-floxed GFP (mT/mG) mouse (Jackson Labs; 007676). Female double-transgenic hypoxia fate-mapping mice were bred to an FVB/N-Tg(MMTV-PyMT) (Jackson Labs; 002374) male mouse to generate triple-transgenic female mice that develop mammary tumors comprising cells that

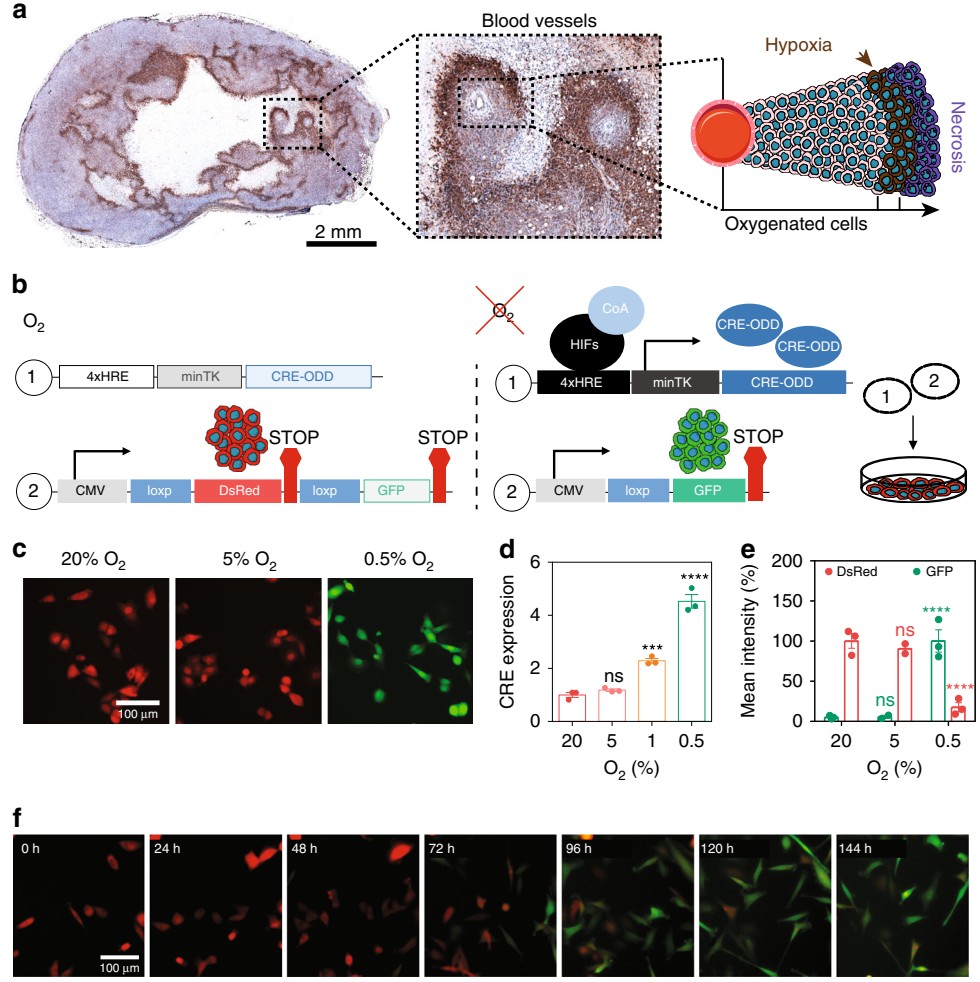

**Fig. 1** Establishing a hypoxia fate-mapping system. **a** A full cross section of a mouse mammary tumor stained with Hypoxyprobe™ (brown). The inset shows cancer cells surrounding two blood vessels. The cartoon depicts the spatial distribution of hypoxic cells (brown) adjacent to necrotic cells (purple) and oxygenated cells (pink). **b** Two lentiviral vectors used to generate a hypoxia fate-mapping system. **c** Fluorescent images of MDA-MB-231 hypoxia fate-mapping cells following exposure to 20%, 5%, or 0.5% $O_2$ for 7 days. Scale bar: 100 μm. **d** Relative expression of CRE mRNA levels measured by qPCR in hypoxia fate-mapping MDA-MB-231 cells exposed to 20%, 5%, 1%, or 0.5% $O_2$ (mean ± SEM, $N = 1$, $n = 3$); ****$P < 0.0001$ versus 20% (one-way ANOVA with Bonferroni posttest). **e** Normalized quantification of flow cytometry analysis of MDA-MB-231 hypoxia fate-mapping cells following exposure to 20%, 5%, or 0.5% $O_2$ for 7 days (mean ± SEM, $N = 2$–3, $n > 10,000$ for 20%, 5%, and 0.5% $O_2$); ****$P < 0.0001$ versus 20% DsRed or GFP (Two-way ANOVA with Bonferroni posttest). **f** Fluorescent live-cell time-lapse imaging of MDA-MB-231 hypoxia fate-mapping cells cultured under 0.5% $O_2$ imaged over a 6-day time course (see also Supplementary movie 1)

switch from tdTomato (tdTom) to permanent GFP expression under hypoxic conditions (Fig. 2a, b). We systematically studied hypoxia during each stage of breast cancer progression:[33] ductal hyperplasia, ductal carcinoma in situ (DCIS), early carcinoma, and late-stage invasive carcinoma (Fig. 2c). Intratumoral hypoxia was first detected in DCIS lesions (Fig. 2c and Supplementary Fig. 2c). We confirmed the localization of hypoxia by Hypoxyprobe™ labeling, which overlapped with GFP+ areas in MMTV-PyMT tumors (Supplementary Fig. 2d).

**Fate-mapping hypoxic cells in 3D models**. To explore the utility of our system to map cells that experience low $O_2$ concentrations in 3D systems in vitro, we generated (Fig. 3a) and imaged spheroids derived from MDA-MB-231 or MCF7 fate-mapping cells fully embedded in a 3D collagen matrix in real time. Unexpectedly, MCF7 spheroids, cultured under 20% $O_2$, displayed regions of hypoxia (GFP+) beginning on day 10, whereas spheroids cultured under 1% $O_2$ contained detectable GFP+ regions on day 2 (Supplementary Fig. 3a, b). A 3D MDA-MB-231

spheroid reconstruction displayed a compact core of GFP+ cells following 15 days of culture under 20% $O_2$. Cells exposed to hypoxia migrated away from the core of the spheroid, where they become hypoxic, to the more oxygenated periphery of the spheroid (Fig. 3b, Supplementary Fig. 3c and Supplementary Movie 2). The size of the GFP+ core increased over time comprising more than half of the spheroid radius on day 20 (Fig. 3c).

To directly measure the $O_2$ gradients in a spheroid cultured under 20% $O_2$, we utilized a retractable-needle fiber-optic microprobe and determined that the core of the spheroid contained ~1% $O_2$ as compared with ~11% $O_2$ measured within the surrounding collagen matrix (Fig. 3d). We also used a fluorogenic compound (Image-iT™ Hypoxia Reagent) to detect the nascent hypoxic regions[34]. The image-iT reagent can be used in conjunction with our system to detect cells that are currently hypoxic (Image-iT+ /GFP+) and those that have been previously exposed to hypoxia but that are no longer hypoxic (Image-iT–/GFP+) (Supplementary Fig. 3d). Our results show that (1) cells cultured in a 3D spheroid can be exposed to varying levels of $O_2$ based on their position within the spheroid and time in culture,

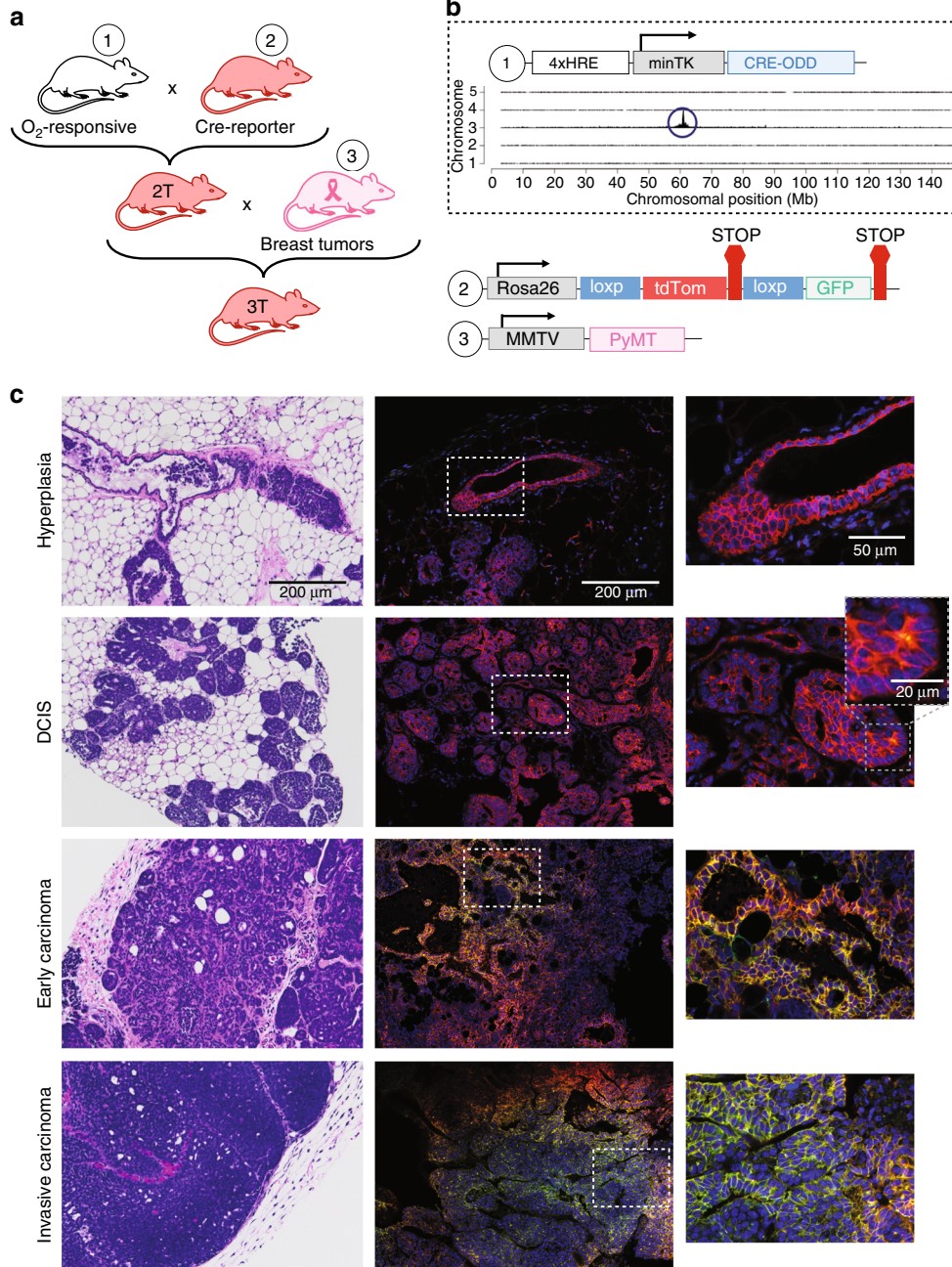

**Fig. 2** Establishing a hypoxia fate-mapping triple-transgenic mouse model. **a** Design of a triple-transgenic mouse generated by (1) developing a transgenic mouse that produces Cre recombinase in cells exposed to hypoxia followed by (2) breeding to the mT/mG reporter mouse (Jackson Labs; 007676). The double-transgenic mouse (2 T) was then crossed to a (3) mouse expressing the MMTV promoter-driven PyMT oncogene that develops spontaneous breast tumors (3 T). **b** TLA sequence coverage across the mouse genome by using primer sets 1 and 2 (Supplementary Table 2) was conducted to determine the location of our transgene. The 4xHRE-CRE-ODD transgene integrated at chromosome 3 position 61,062,240. **c** Triple-transgenic mice were sacrificed at different time points over a 4-month period in order to detect hypoxic cells during breast cancer progression from hyperplasia, to ductal carcinoma in situ (DCIS), to early carcinoma and invasive late-stage carcinoma. H&E-stained sections of paraffin-embedded tissue (left) or fluorescent imaging of frozen tissue sections (right). Marked insets are displayed on the right. The second inset at the DCIS stage highlights early detection of hypoxia

and (2) GFP+ cells can migrate from the core to more oxygenated regions such as the periphery of the spheroid. To further validate our triple-transgenic mouse system, we generated mammary organoids derived from tdTom+ DCIS lesions and cultured them under 20% or 0.5% $O_2$ (Fig. 3e and Supplementary Fig. 4a, b). The organoids switched from tdTom expression to GFP expression following 10 days of culture under 0.5% $O_2$ conditions (Fig. 3f and Supplementary Fig. 4b). Organoids cultured under 20% $O_2$ showed higher proliferation rates than those cultured under 0.5% $O_2$ conditions and reached variable diameters, up to 0.5 mm (Supplementary Fig. 4c). Larger organoids contained GFP+ regions centrally located within the organoid, which were Image-iT™ positive (Fig. 3f, Supplementary Fig. 4a, f and Supplementary Movie 3). Flow cytometry was performed for confirmation (Fig. 3g and Supplementary Fig. 4d, e). To measure $O_2$ concentrations in the organoid, we incorporated REDFLASH luminescent nanoprobes into the 3D Matrigel matrix surrounding embedded organoids. $O_2$ levels were

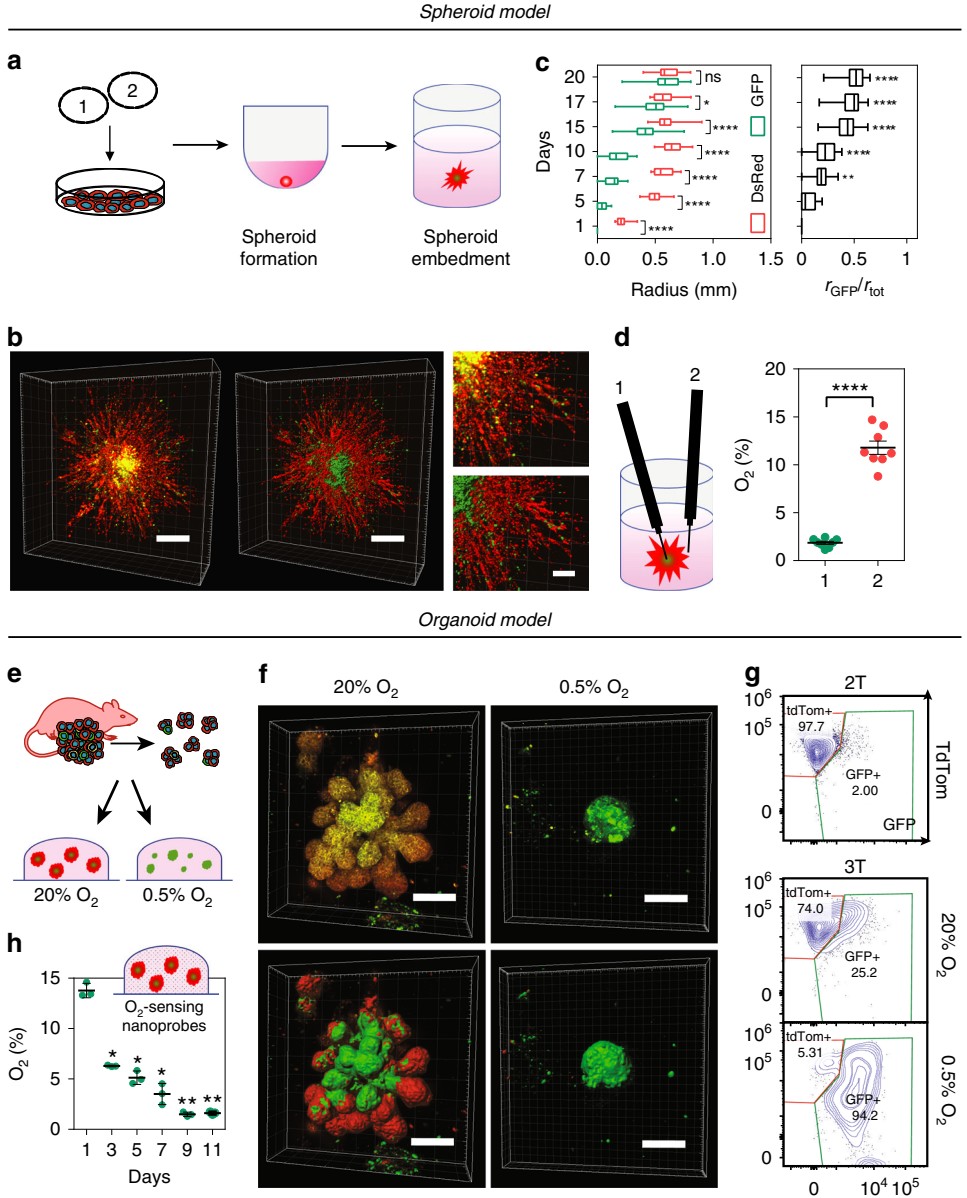

**Fig. 3** Fate mapping of hypoxic cells in 3D models. **a** Hypoxia fate-mapping cells were used to generate 3D spheroid cultures. Spheroids were formed in spheroid formation media in round-bottom well plates and transferred to the center of 3D collagen matrices 72 h later. **b** 3D reconstruction of a spheroid imaged after 15 days in culture and correspondent 3D surface rendering (right and bottom). Scale bar: 500 μm (left), 200 μm (right). **c** Analysis of color distribution measured along the spheroid radius (mean ± SEM, $N = 3$, $n = 27$); ****$P < 0.0001$ GFP versus DsRed on each day (Two-way ANOVA with Bonferroni posttest) (left). Length of the GFP+ radius ($r_{GFP}$) over the total spheroid radius ($r_{tot}$) (mean ± SEM, $N = 3$, $n = 27$); ****$P < 0.0001$ versus day 5 (one-way ANOVA with Bonferroni posttest) (right). The box extends from the 25th to 75th percentiles, the median is marked by the vertical line inside the box, and the whiskers represent the minimum and maximum points. **d** $O_2$ measurements in the core (1) or periphery (2) of MDA-MB-231 spheroids following 20 days in culture under 20% $O_2$ (mean ± SEM, $N = 8$); ****$P < 0.0001$ position 1 versus 2 (two-tailed $t$-test). **e** Organoids derived from 3 T mouse tumors were embedded in 3D in Matrigel and cultured under 20% or 0.5% $O_2$. **f** 3D reconstruction of fluorescent images of organoids following 10 days of culture under 20% or 0.5% $O_2$ with corresponding 3D surface rendering (bottom). Scale bar: 100 μm. **g** Representative contour plots of flow cytometry data from 3 T mouse tumor organoids cultured under 20% or 0.5% $O_2$ for 20 days. 2 T tumor organoids were used to define tdTom+ and GFP+ gates for flow cytometry analysis (Supplementary Fig. 4d). **h** $O_2$ measurements performed by using OXNANO nanoprobes dispersed in 3D Matrigel surrounding embedded organoids (mean ± SEM, $N = 3$, $n > 200$); ****$P < 0.0001$ versus day 1 (matched one-way ANOVA with Bonferroni posttest)

14% 24 h after embedding into Matrigel and dropped to 1.7% $O_2$ after 11 days in culture (Fig. 3h). The expression levels of P4HA1, a well-established $O_2$-regulated gene[35], and GFP were assessed by qPCR and are consistent with $O_2$ measurements (Supplementary Fig. 4g, h). Taken together, the results suggest that (1) organoids cultured in 3D develop severe $O_2$ gradients that range from 14% to 1.7%; (2) the change in $O_2$ levels over time may be an

important environmental cue that dictates the fate of individual or clusters of cells in organoids grown in 3D cultures.

**Fate-mapping intratumoral hypoxia.** We generated orthotopic breast tumors by transplanting MDA-MB-231 (or 4T1) fate-mapping cells into the mammary fat pad of mice followed by

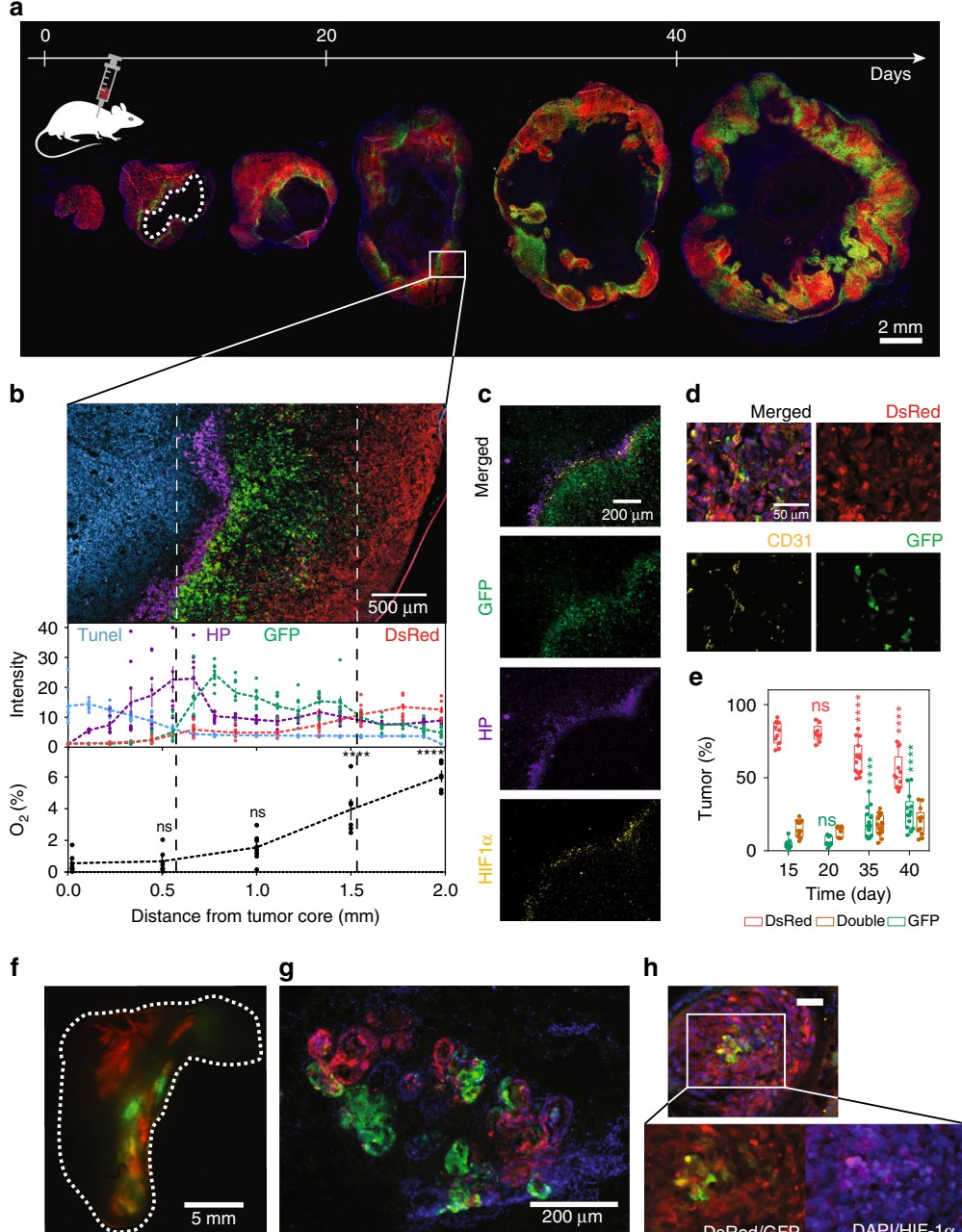

**Fig. 4** Implementing a hypoxia fate-mapping system in breast cancer models. **a** Fluorescent images of full cross sections of orthotopic mammary tumors derived from MDA-MB-231 hypoxia fate-mapping cells. Tumors were excised at various times over a 45-day period and sectioned for imaging. The white dashed line outlines the necrotic core. **b** Fluorescent images of DsRed and GFP expression, and TUNEL (blue) and Hypoxyprobe™ (purple) labeling. Normalized intensity plots for each fluorophore (top) (mean ± SEM, $n = 8$), or $O_2$ measurements (bottom) are displayed as a function of distance from the tumor core. $O_2$ measurements were performed by using a fixed-needle microprobe (mean ± SEM, $N = 8$; ****$P < 0.0001$ vs. 0 (one-way ANOVA with Bonferroni posttest). **c** Fluorescent images of GFP expression and Hypoxyprobe™ (purple) and HIF-1α (yellow) immunofluorescent labeling. **d** Fluorescent images of DsRed and GFP expression, and CD31 (yellow) immunofluorescent labeling to detect endothelial cells lining blood vessels in a tumor region far from the necrotic core. **e** Full tumor cross sections were imaged in tiles, linearly stitched, binarized by ImageJ, and used to determine the ratio of DsRed-, double- (both DsRed and GFP), and GFP-positive areas of MDA-MB-231 orthotopic tumors (see Supplementary Fig. 5b for an illustration of the calculation method). Ratios (%) are displayed in the graph at different time points of tumor progression (mean ± SEM, $N = 3$, $n = 48$); ****$P < 0.0001$ versus day 15 (Two-way ANOVA with Bonferroni posttest). The box extends from the 25th to 75th percentiles, the median is marked by the vertical line inside the box, and the whiskers represent the minimum and maximum points. **f** MCF7 hypoxia fate-mapping cells were injected into the nipple and delivered to a single ductal tree of multiparous NSG mice. A whole mount of the mammary fat pad was imaged by fluorescent microscopy 60 days after injection. **g** Tissue sections were stained with DAPI to detect cell nuclei and imaged for DsRed and GFP. **h** Tissue sections were stained with DAPI to detect cell nuclei and labeled with a HIF-1α antibody. Scale bar: 50 μm

tumor resection at different time points (Fig. 4a and Supplementary Fig. 5a). To correlate fluorescence expression with intratumoral $O_2$ concentrations, we directly measured percent $O_2$ as a function of position within the tumor by using a fixed-needle microprobe, demonstrating that cells express GFP when they experience <0.5% $O_2$ in vivo (Fig. 4b and Supplementary Fig. 5b). Hypoxyprobe™ labeled cells were localized in the perinecrotic (TUNEL+) region of the tumor and co-localized with GFP+ (Fig. 4b and Supplementary Fig. 6a) and HIF-1α staining (Fig. 4c). GFP+ cells intermingle with blood vessels and reach the invasive front of the tumor (Fig. 4d and Supplementary Fig. 6b), suggesting that a portion of hypoxic cells have the ability to migrate from the perinecrotic region of the tumor into more oxygenated tissue regions. The distribution of GFP+ to DsRed+ cells in the primary tumor increased over time (Fig. 4e and Supplementary Fig. 5c).

Since 70–75% of all breast cancers are estrogen receptor α-positive (ER+)[36,37], we established an ER+ hypoxia fate-mapping model by using MCF7 cells. MCF7 hypoxia fate-mapping cells were injected directly into the nipple and delivered to a single ductal tree of multiparous NSG mice, avoiding the requirement for estrogen supplementation[38]. Sixty days after MCF7 cell injection, many of the mammary ducts became occluded, displayed regions of hypoxia (GFP+), and disrupted the basal membrane of the duct wall (Fig. 4f, g). GFP+ cells were positive for HIF-1α staining (Fig. 4h).

Together, these data show that cells that experienced hypoxia in the primary tumor (GFP+) can migrate toward a more oxygenated invasive front of tumor regions in orthotopic MDA-MB-231 and 4T1 models and form invasive structures in the intraductal MCF7 model.

**Intratumoral hypoxia profiling serves as a prognostic tool**. The standard method for studying gene expression changes that occur under hypoxic conditions in vitro involves exposing cells to 20% compared with 1% $O_2$ for short periods of time (on the order of hours to <3 days), although 20% $O_2$ is substantially higher than the normal $O_2$ levels in even well-perfused tissues such as the lung parenchyma[39]. Therefore, we compared the gene expression changes that occur in vivo with standard hypoxic exposure in vitro.

To ensure that the GFP+ cells were hypoxic at the time of recovery, we sorted GFP+ and DsRed+ cancer cells from 3-mm-size tumors and confirmed Hypoxyprobe™ expression in the GFP+ population. RNAsequencing of the GFP+ (TG) and DsRed+ (TR) cells revealed 214 genes that were upregulated (FC ≥ 1.5) in GFP+ when compared with DsRed+ cells, and 49 genes were downregulated (FC ≤ −1.5) (Supplementary Fig. 7a–c and Supplementary table 4). By using gene set enrichment analysis (GSEA)[40,41], we determined that GFP+ tumor cells were enriched for the Hallmark gene set of hypoxia (Supplementary Fig. 7d, e and Supplementary Table 5). In contrast, DsRed+ tumor cells were enriched for Hallmark gene sets related to mTOR, Myc, E2F, and oxidative phosphorylation signaling pathways, among others, which are known to be repressed by hypoxia[42–46] (Supplementary Fig. 7d and Supplementary Table 5).

We then compared gene expression changes caused by intratumoral hypoxia with changes that occur in MDA-MB-231 cells exposed to hypoxia in vitro. For in vitro studies, we exposed cells to 20% or 1% $O_2$ for 24 h, followed by RNA-sequencing. Forty-one RNA transcripts were regulated in common in GFP+ versus DsRed+ cancer cells sorted from primary tumors and MDA-MB-231 cells exposed to 1% $O_2$ versus 20% $O_2$ in vitro (Fig. 5a and Supplementary Table 6). qPCR of independent RNA

samples was used to determine the expression level of CRE, a readout for the activation of our system (Fig. 5b), and the HIF-inducible genes[16] CA9, DNAH11, EGLN3, and LOX (Fig. 5c). The basal level of CRE, CA9, DNAH11, EGLN3, and LOX in DsRed+ cells was 3- to 100-fold greater than that in cells cultured under 20% $O_2$ in vitro. The results correspond to our finding that DsRed+ tumor cells in vivo experience $O_2$ levels closer to 2–6% (Fig. 4b), which is also consistent with HIF protein levels measured under different $O_2$ levels[47]. Interestingly, almost 800 genes were upregulated upon exposure to hypoxia in vitro but not in vivo. Genes that were upregulated exclusively in GFP+ versus DsRed+ cancer cells but not by exposure to hypoxia in vitro did not follow the same pattern of regulation as the overlapping gene set. For example, CP and ITGA10 were upregulated exclusively in GFP+ versus DsRed+ tumor cells but not at all by in vitro exposure to hypoxia (Fig. 5d). A heat map of the 41-gene signature derived from the overlap of intratumoral and in vitro hypoxia revealed a correlation ($r = 0.84$) between the distribution of the relative fold change of each gene in GFP+ versus DsRed+ tumor cells (T) and cells exposed to 1% versus 20% $O_2$ (C) (Fig. 5e).

In order to determine whether the response to hypoxia is reversible, we generated tumor-derived cell lines from hypoxia fate-mapping tumors by sorting DsRed+ and GFP+ cancer cells directly into culture media (Supplementary Fig. 8a). Cells were cultured and collected after ~20 cell doublings (3–5 passages) to assess CRE, CA9, DNAH11, EGLN3, and LOX expression levels by qPCR. After reoxygenation and in vitro cell culture, GFP+ cells retained increased expression in some but not all genes (Supplementary Fig. 8b, c). In parallel studies, we cultured cells under hypoxia in vitro for 5 days followed by 48-h days of reoxygenation under 20% $O_2$ (Supplementary Fig. 8d). CRE, CA9, EGLN3, DNAH11, and LOX expression returned to pre-exposure levels (Supplementary Fig. 8e, f).

Together, these data demonstrate that (1) the response to in vitro hypoxic culture conditions (24 h) does not recapitulate the response to physiologic levels of intratumoral hypoxia; (2) intratumoral hypoxia-induced gene expression is not entirely reversible even after standard tissue culture, whereas the in vitro hypoxic effect is reversed within 2 days of culture under 20% $O_2$.

To determine the prognostic potential of the 3 differentially expressed gene sets that we identified (Fig. 5a), we analyzed microarray expression data derived from breast cancer tissue[48]. We compared the 41-gene signature derived from the overlap of intratumoral and in vitro hypoxia exposure, as well as the 40 most differentially expressed genes in the intratumoral hypoxia or in vitro hypoxia subgroups. Patients ($N = 664$) were stratified as either having high or low expression of each gene signature by using median expression as the cutoff. The 41-gene signature derived from the overlap of intratumoral and in vitro hypoxia exposure had the most significant prognostic potential (Fig. 5f–h and Supplementary Fig. 7f).

To further explore the observed differences between intratumoral and in vitro exposure to hypoxia, we used the GSEA[40,41] pre-ranked analysis tool to investigate potential differences in the expression of Hallmark gene sets (Supplementary Fig. 7g). We compared matched gene lists by ranking each gene identified by RNA sequencing according to fold change upon exposure to hypoxia. In vitro exposure to hypoxia showed higher enrichment in the hallmarks hypoxia, EMT, and glycolysis, well-described hypoxia-regulated pathways[49,50]. Hedgehog signaling and TNFα were significantly enriched in vivo but not in vitro[51,52]. Both in vitro and intratumoral hypoxia showed significant dow-regulation of pathways associated with proliferation.

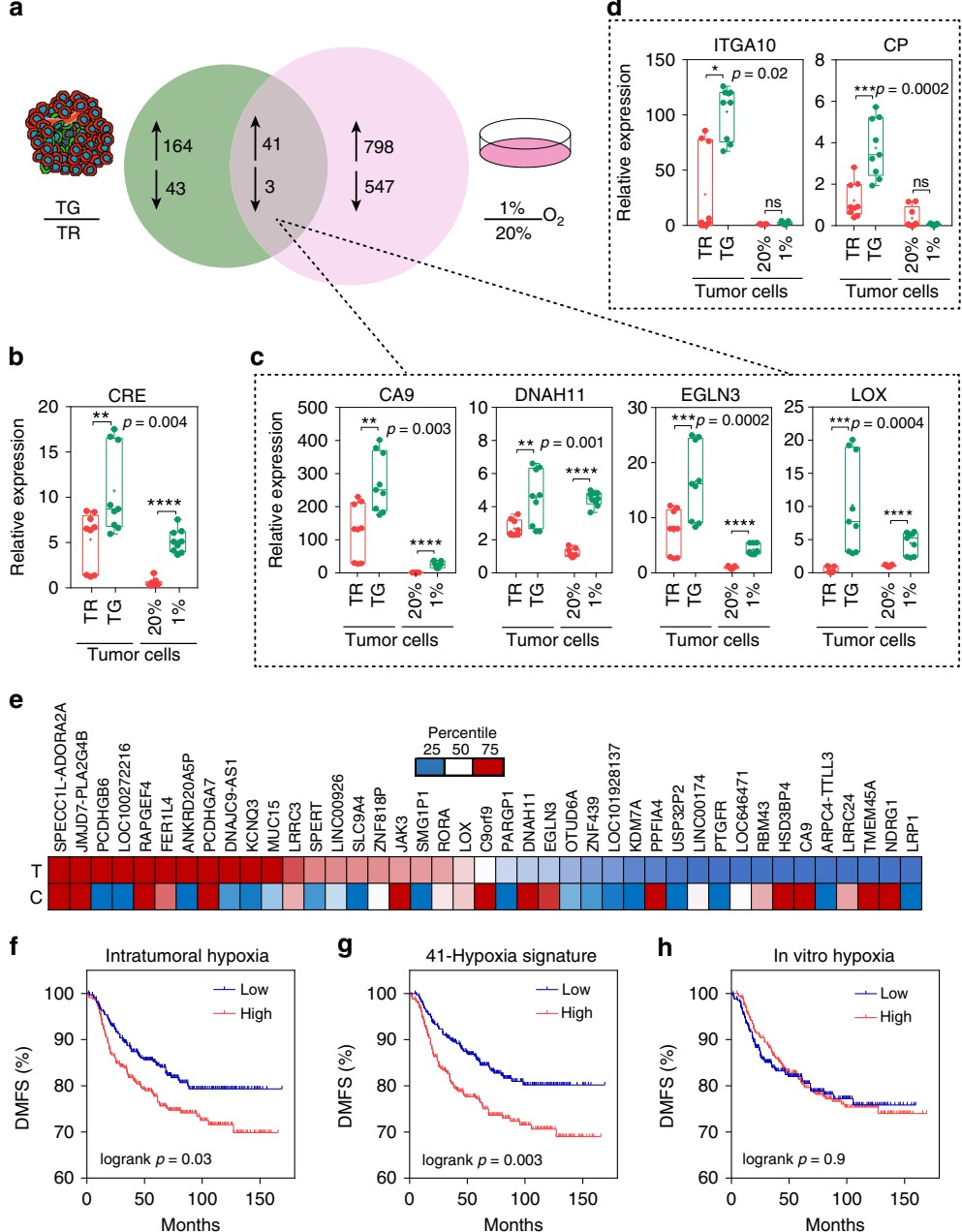

**Fig. 5** Hypoxia fate-mapping system facilitates profiling of intratumoral hypoxia. **a** Tumors derived from hypoxia fate-mapping MDA-MB-231 cells were harvested 2 weeks after implantation and sorted into DsRed+/GFP− or GFP+ populations directly into Trizol (N = 2). Venn diagram displaying the overlap of the number of genes with differential expression (−1.5 ≥ FC ≥ 1.5) in GFP+ versus DsRed+ tumor cells (green circle) and MDA-MB-231 cells exposed to 1% $O_2$ versus 20% $O_2$ (pink circle) (Supplementary Table 6). **b–d** Relative expression of mRNA measured by qPCR in tumor-derived cells (TG or TR) or MDA-MB-231 cells exposed to 20% or 1% $O_2$ in vitro, **b** CRE, **c** genes co-regulated by intratumoral and in vitro hypoxia (CA9, DNAH11, EGLN3, and LOX), and **d** genes exclusive to upregulation upon intratumoral hypoxia (ITGA10 and CP) (mean ± SEM, N = 3, n = 3); ****P < 0.0001 TG versus TR and 1% versus 20% (two-tailed t-test). The box extends from the 25th to 75th percentiles, the median is marked by the vertical line inside the box, and the whiskers represent the minimum and maximum points. **e** Heat map of the 41-gene signature derived from the overlap of intratumoral and in vitro hypoxia. The distribution of the relative fold change of each gene in the 41-gene signature is displayed for GFP+ versus DsRed+ sorted tumor cells (T) or cells (C) exposed to 1% versus 20% $O_2$ conditions. Genes with fold change higher than 75% of the fold change of genes in the set are red and genes with fold change lower than 25% of the fold change of genes in the set are blue (Pearson correlation factor r = 0.84 and P = 9.6 × 10$^{-7}$). **f–h** Microarray expression data from 664 breast cancer patients were used to perform multigene survival analysis (n = 664; HGU133 plus 2.0 arrays, KMplotter). Kaplan–Meier analysis of distant metastasis-free survival (DMFS) of breast cancer patients stratified by high or low expression by using **f** the 40 most induced genes by intratumoral hypoxia, **g** the 41-gene signature derived from the overlap of intratumoral and in vitro hypoxia, or **h** the 40 most induced genes by in vitro hypoxia

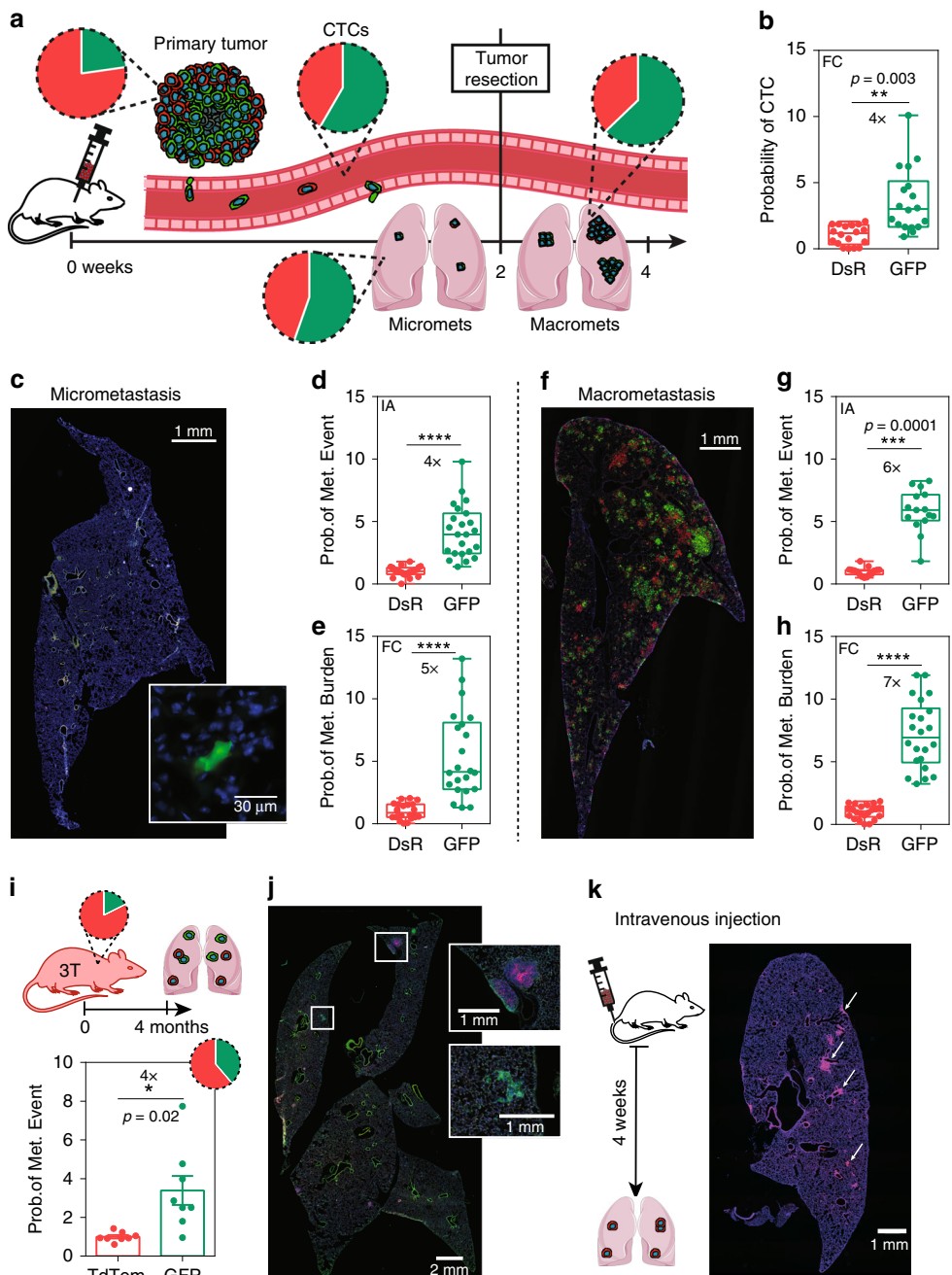

**Fig. 6** Post-hypoxic cells have enhanced metastatic potential. **a** Orthotopic tumors derived from MDA-MB-231 hypoxia fate-mapping cells. Blood, lungs, and tumor were harvested at 2 weeks for half of the mouse cohort. Surgical tumor resection was performed on the second half of the cohort, and lungs were harvested 2 weeks after tumor removal to assess late-stage metastasis. Pie charts represent the ratio of DsRed+ to GFP+ cells in each site. **b** The probability of a GFP+ (or DsRed+) CTC in the blood was obtained by dividing the percentage of GFP+ (or DsRed+) CTCs detected in the bloodstream with the percentage of GFP+ (or DsRed+) cells in the matched primary tumor ($N = 3$, $n = 17$) by using flow cytometry (FC); ****$P < 0.0001$ GFP versus DsRed (two-tailed $t$-test). **c** Cross-sectional imaging of DsRed and GFP expression and DAPI labeling of lung micrometastasis at week 2. The inset shows magnified GFP+ micrometastasis. **d**, **e** The probability of a GFP+ metastatic event in the lung is obtained by dividing the percentage of GFP+ micrometastases by the percentage of GFP+ cells in the matched primary tumor by using **d** image analysis or **e** flow cytometry ($N = 3$, IA: $n = 23$; FC: $n = 22$); ****$P < 0.0001$ GFP versus DsRed (two-tailed $t$-test). The box extends from the 25th to 75th percentiles, median is marked by the line inside the box, and whiskers represent the minimum and maximum. **f** Cross-sectional imaging of DsRed and GFP expression and DAPI labeling of lung macrometastasis 2 weeks after tumor resection. **g**, **h** The probability of a GFP+ metastatic event in the lung is obtained by dividing the percentage of individual GFP+ macrometastatic colonies by the percentage of GFP+ cells in the matched primary tumor by using **g** image analysis or **h** flow cytometry ($N = 3$, IA: $n = 16$; FC: $n = 22$); ****$P < 0.0001$ GFP versus DsRed (two-tailed $t$-test). **i**, **j** Tumor and lungs from triple-transgenic mice were harvested at 4 months of age. Probability of lung metastatic events was obtained as described in **g** (mean ± SEM, $N = 8$); ****$P < 0.0001$ GFP versus tdTom (two-tailed $t$-test). **k** MDA-MB-231 hypoxia fate-mapping cells were injected into the tail vein of NSG mice and harvested 5 weeks later

**Post-hypoxic cells have enhanced metastatic potential.** To investigate whether cells that become hypoxic in the perinecrotic region of primary tumors (Fig. 4a) have the ability to metastasize to distant organs, we removed 3-mm-size tumors from mice bearing hypoxia fate-mapping MDA-MB-231 tumors and harvested matched blood and lungs to assess the ratio of DsRed+ to GFP+ cells either at the time of tumor removal or 2 weeks later (Fig. 6a). At the time of tumor removal, <25% of cells in the primary tumor were GFP+, yet DsRed+ and GFP+ cells were found in similar numbers in the bloodstream. The results suggest that cells that experience hypoxia have a fourfold increased probability of accessing the bloodstream as compared with DsRed+ cells (Fig. 6b and Supplementary Fig. 9a, b). GFP+ cells maintained a four- to fivefold increased probability in the ability to seed the lung as single or aggregated cells (Fig. 6c–e).

Three weeks following tumor removal, we assessed the ability of GFP+ cells to form overt macrometastasis. GFP+ cells were 6–7× more likely than DsRed+ cells to form lung metastasis (Fig. 6f–h). In parallel studies, we assessed lung metastasis in 8 triple-transgenic FVB mice as well as the syngeneic 4T1 model, and observed similar metastatic capabilities in an immune-competent background (Fig. 6i, j and Supplementary Fig. 9c–f).

To be certain that the DsRed to GFP switch does not occur in the circulation or after arrival in distant organs, we directly injected the FAC-sorted DsRed+ cells into the tail vein of mice. The cells remained DsRed+ upon colonization of the lung (Fig. 6k), reflecting the higher $O_2$ levels in this organ as compared with intratumoral regions[53]. This finding confirms that DsRed to GFP switch occurs in the primary tumor and not in the circulation or in the lung. We also FAC-sorted DsRed+ cells from the tumor and confirmed that they retain the ability to express GFP when exposed to 0.5% $O_2$ in vitro (Supplementary Fig. 9g).

Together, the findings demonstrate that (1) a portion of hypoxic cells in regions adjacent to the perinecrotic region of the tumor can overcome $O_2$ deprivation and metastasize; (2) cells that leave the hypoxic region of the primary tumor and enter the bloodstream will be exposed to higher $O_2$ levels (reoxygenation); (3) GFP+ cells detected in the blood and lung became hypoxic in the primary tumor; (4) post-hypoxic cells have 4–5× higher chance at being found in the blood and lung.

**Exposure to intratumoral hypoxia promotes invasion.** The fact that viable GFP+ cells have a 4× increased probability to be found in the blood than viable DsRed+ cells suggests that they are better equipped to accomplish the earliest steps in the metastatic cascade, which include local invasion, intravasation, and survival in the bloodstream (Fig. 7a). To compare the invasive potential of GFP+ and DsRed+ cells, we isolated cancer cells from the primary tumor and incorporated them with 90% non-labeled MDA-MB-231 cells to generate spheroids to track cell migration (Fig. 7b). Cell trajectories were fit by using an anisotropic persistent random walk (APRW) model[54], and GFP+ cells had a moderate but significant increase in total diffusivity and persistent time, as well as longer projections of cell trajectories. GFP+ cells were more often localized at the invasive front of the spheroid (Fig. 7c–f). GFP+ tumor cells were enriched for gene sets related to motor proteins, such as myosin, dynein, and actin, and cellular motor projections such as cilia and microvillus, compared with DsRed+ cells (Supplementary Fig. 10a). GFP+ cells at the tumor periphery displayed projections, suggesting the presence of protrusions that potentially contribute to enhanced invasion (Supplementary Fig. 10b).

**Post-hypoxic CTCs have a ROS-resistant phenotype.** Invasion is not the rate-limiting step in metastasis as thousands of CTCs are shed per day in patients with aggressive tumors, and only a small fraction of them contribute to metastasis[55]. Therefore, we reasoned that in addition to enhanced invasion, GFP+ cells might be primed to have a survival advantage in the bloodstream. Recent work has shown that oxidative stress is a limiting step in the metastatic process because it prevents the survival of CTCs in the bloodstream[56], so we questioned whether GFP+ cells were more resistant to oxidative stress (Fig. 8a). DsRed+ cells in the primary tumor are enriched for the reactive oxygen species (ROS), fatty acid metabolism, and oxidative phosphorylation pathways that are known to contribute to ROS (Supplementary Fig. 7d).

In order to evaluate mitochondrial ROS levels directly, we isolated GFP+ and DsRed+ cancer cells in the primary tumor and blood in mice and stained for MitoROS. The levels of ROS in DsRed+ cells were significantly higher than GFP+ cells in both the primary tumor and in the blood. DsRed+ cells had a 2-fold increase in ROS levels in the blood as compared with the primary tumor, whereas GFP+ cells maintained the same level of ROS in both the tumor and bloodstream (Fig. 8b). By treating freshly harvested CTCs with the ROS inhibitor N-acetylcysteine (NAC), ROS levels in DsRed+ CTCs were reduced to ROS levels measured in GFP+ cells (Fig. 8c). The increased level of ROS correlates with the decreased viability of DsRed+ CTCs freshly harvested from the bloodstream (Fig. 8d). To confirm this ROS-resistant phenotype, we treated GFP+ or DsRed+ cells derived from primary tumors with hydrogen peroxide ($H_2O_2$) for 1 hour to promote ROS-induced apoptosis and evaluated survival 48 h later. GFP+ cells were significantly more resistant to $H_2O_2$ (50 and 500 μM) than DsRed+ cells (Fig. 8e, f and Supplementary Fig. 11a). In parallel studies, we also tested the survival of GFP+ and DsRed+ cells when cultured in a suspension assay or when challenged with shear stress, to simulate the dynamics that cells encounter in the bloodstream. In both assays, GFP+ cells were significantly more resistant to cell death (Supplementary Fig. 11b, c). Together, the findings demonstrate that GFP+ cells are more resistant to oxidative stress and maintain lower levels of mitochondrial ROS in the bloodstream than DsRed+ cells. DsRed+ cells are also more susceptible to ROS-induced apoptosis.

To determine whether the ROS-resistant phenotype could enhance the ability of GFP+ cells to accomplish late steps of metastasis—extravasation and colonization, we injected equal numbers of GFP+ and DsRed+ cells isolated from primary tumors into the tail vein of mice (Fig. 8g). Similar numbers of DsRed+ and GFP+ cells were found in the lung 48 hours after intravenous injection (Fig. 8h, j, k). Whereas, lungs harvested 25 days after tail-vein injection contained 2× more GFP+ colonies than DsRed+ cells (Fig. 8i–k). DsRed+ and GFP+ cells formed similar-size metastatic lung nodules (Supplementary Fig. 11d) suggesting that post-hypoxic cells do not have a proliferation advantage at the metastatic site. Two times more GFP+ than DsRed+ cells that successfully extravasate are able to initiate metastatic colonization. Taken together, our results suggest that post-hypoxic cells acquire a ROS-resistant phenotype in the primary tumor, as well as the ability to migrate and intravasate into the bloodstream. The ROS-resistant phenotype allows them to survive high ROS levels in the bloodstream to promote their metastasis-initiating capability in the lung.

**Post-hypoxic 'memory' at the metastatic site.** In order to determine whether the ROS-resistant phenotype would persist in cells that successfully colonized the lung, we performed RNA-sequencing analysis. One-hundred and twenty-one genes were upregulated (FC ≥ 1.5) in GFP+ compared with DsRed+ metastatic cells, and 47 genes were downregulated (FC ≤ −1.5)

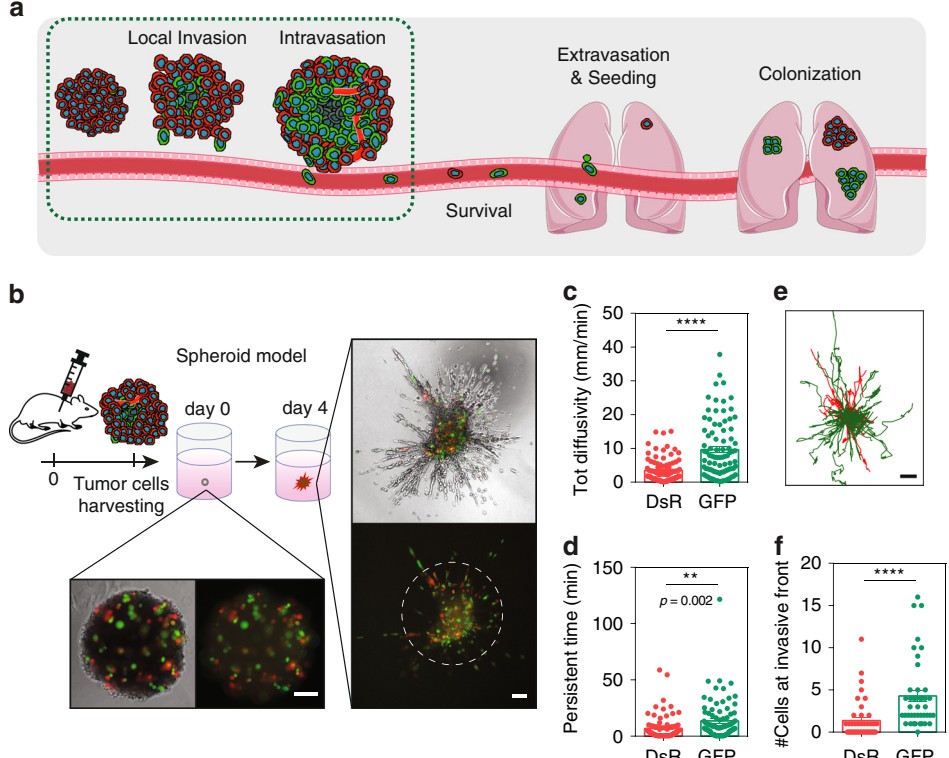

**Fig. 7** Exposure to intratumoral hypoxia promotes invasion. **a** Schematic representation of the early and late steps of the metastatic cascade. **b** Tumor-derived cells were incorporated with 90% non-labeled wild-type MDA-MB-231 cells in a 3D spheroid fully embedded in collagen. After 4 days in culture, spheroids were subjected to time-lapse imaging every 5 min for 16 h. Phase-contrast and fluorescent images taken on days 0 and 4 of the same spheroid. Scale bar = 100 μm. The white dashed ring marks the invasive front of the spheroid. **c, d** Total diffusivity (mm/min) (**c**) and persistent time (min) (**d**) of DsRed+ and GFP+ (mean ± SEM, N = 3, n = 80–92 cells); ****P < 0.0001 GFP versus DsRed (two-tailed t-test). **e** Projection of DsRed+ (red) or GFP+ (green) cell trajectories. Scale bar = 100 μm. **f** Number of DsRed+ and GFP+ cells at the invasive front of the spheroid (beyond the dashed white line in (**b**)) (mean ± SEM, N = 3, n = 44 spheroids); ****P < 0.0001 GFP versus DsRed (two-tailed t-test)

(Supplementary Fig. 12a, b and Supplementary table 7). The Hallmark ROS pathway was no longer enriched in DsRed+ cells (Supplementary Fig. 12c and Supplementary Table 8).

To determine which genes were upregulated on exposure to intratumoral hypoxia and later in GFP+ compared with DsRed+ lung metastasis, we compared both gene expression profiles. Nineteen RNA transcripts were regulated in common in tumor and lung metastatic GFP+ versus DsRed+ cells (Fig. 9a and Supplementary table 9). We confirmed the expression of CA9, CP, DNAH11, and EGLN3 by using independent RNA samples (Fig. 9b). Interestingly, 9 of the 19 genes are known to be regulated in a hypoxia-regulated and HIF-dependent manner (Fig. 9c). By comparing the expression of GFP+ versus DsRed+ cells in the tumor versus the lung, the fold changes were consistent between primary tumor and lung metastasis (r = 0.85, p < 0.0001). Taken together, the data show that a set of genes induced by hypoxia in the primary tumor remain overexpressed at the metastatic site. Whether and how this 'hypoxic memory' occurs is unknown, but it is tempting to speculate that these genes can be used as biomarkers to identify cells at metastatic sites that have been exposed to intratumoral hypoxia.

## Discussion

Clinical studies measuring the PO₂ in breast tumors have shown that patients with tumors containing PO₂ levels <10 mmHg have a worse prognosis[9,12]. While many studies, including our own, have presented data that are consistent with this model, direct evidence that hypoxic cells can metastasize to distant organs is lacking. Previous studies have investigated the ability of hypoxic cells to metastasize by exposing cells to hypoxia in vitro followed by tail-vein injection[23,57,58]. While insightful, this approach lacks the selection that occurs in the primary tumor and also bypasses the migration and intravasation obstacles that are required for metastasis[59]. In addition, this experimental setup does not account for the role of hypoxia in alterations of the primary tumor microenvironment that include promoting angiogenesis[2] and collagen biogenesis[60,61] that drive tumor cell dissemination, as well as regulation of the pre-metastatic niche[62]. Given the experimental evidence to date, it remained possible that hypoxia dictates metastatic outcome by modulating changes inherent only to cells in the primary tumor. For example, hypoxic cells could endow adjacent, but non-hypoxic, cells to metastasize. To address this question, we utilized an in vivo approach allowing cancer cells to be exposed to physiological O₂ levels and exposure times. Our results demonstrate that cells exposed to physiological levels of hypoxia in the primary tumor (GFP+) have a 4× greater probability of becoming a viable CTC than DsRed+ cells. Moreover, post-hypoxic cells have a 6–7× greater probability of forming overt lung metastases suggesting that they have an enhanced metastasis-initiating capability. The enhanced metastatic potency of post-hypoxic cells arises, in part, from a ROS-resistant phenotype that is conferred upon exposure to intratumoral hypoxia. Our data suggest that the portion of perinecrotic cells that can survive, proliferate, migrate to blood vessels, and metastasize are also cells that are selected to be the most resistant to ROS (Fig. 9d).

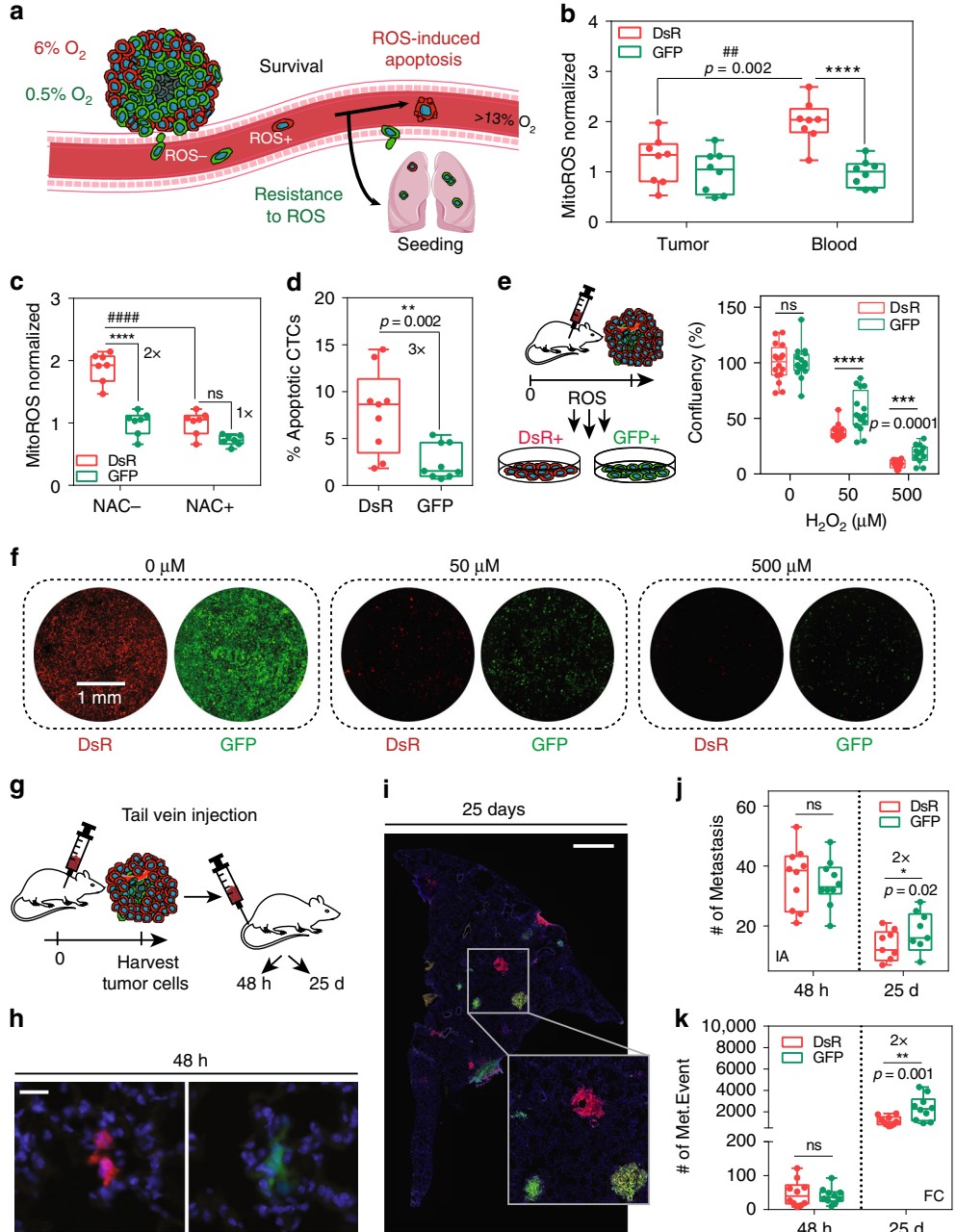

**Fig. 8** Post-hypoxic CTCs have ROS-resistant phenotype. **a** Schematic representation of the proposed mechanism. Post-hypoxic cells (GFP+) are resistant to ROS and have a survival advantage over DsRed+ cells in the bloodstream. **b** Mitochondrial ROS levels were measured by using MitoROS by flow cytometry in matched tumor and blood samples freshly harvested from mice ($N = 2$, $n = 8$); ****$P < 0.0001$ GFP versus DsRed and ####$P < 0.0001$ tumor versus blood (Two-way ANOVA with Bonferroni posttest). The box extends from the 25th to 75th percentiles, the median is the vertical line inside the box, and the whiskers represent the minimum and maximum. **c** Freshly isolated CTCs were treated with the ROS inhibitor N-acetylcysteine (NAC) for 1 h, and ROS levels were measured using MitoROS by flow cytometry ($N = 1$, $n = 7$); ****$P < 0.0001$ GFP versus DsRed and ####$P < 0.0001$ NAC– versus NAC+ (Two-way ANOVA with Bonferroni posttest). **d** Cell viability was measured by using Sytox Blue by flow cytometry ($N = 1$, $n = 9$); ****$P < 0.0001$ GFP versus DsRed (two-tailed $t$-test). **e**, **f** Tumor-derived cells sorted for DsRed+ or GFP+ expression were treated ex vivo with $H_2O_2$ for 1 h and **f** were quantified 48 h later by image analysis by using nuclei segmentation to determine cell confluence (**e**) ($N = 4$, $n = 16$); ****$P < 0.0001$ GFP versus DsRed (Two-way ANOVA with Bonferroni posttest). **g** Tumor-derived hypoxia fate-mapping cells ($1 \times 10^5$ cells) were injected directly into the tail vein of NSG mice. Organs were harvested either 48 h or 25 days after injection. **h**, **i** Cryo-sections of the lung were stained with DAPI and imaged for DsRed and GFP to detect micrometastasis after 48 h (**h**) or macrometastasis after 25 days (**i**). Scale bar: 50 μm (**h**), 1 mm (**i**). **j** Quantification of the number of metastatic events by image analysis of mounted lung sections after 48 h or 25 days ($N = 1$, $n = 9$–10); ****$P < 0.0001$ GFP versus DsRed (Two-way ANOVA with Bonferroni posttest). **k** Quantification of metastatic burden by flow cytometry analysis of lungs after 48 h or 25 days ($N = 1$, $n = 10$); ****$P < 0.0001$ GFP versus DsRed (Two-way ANOVA with Bonferroni posttest)

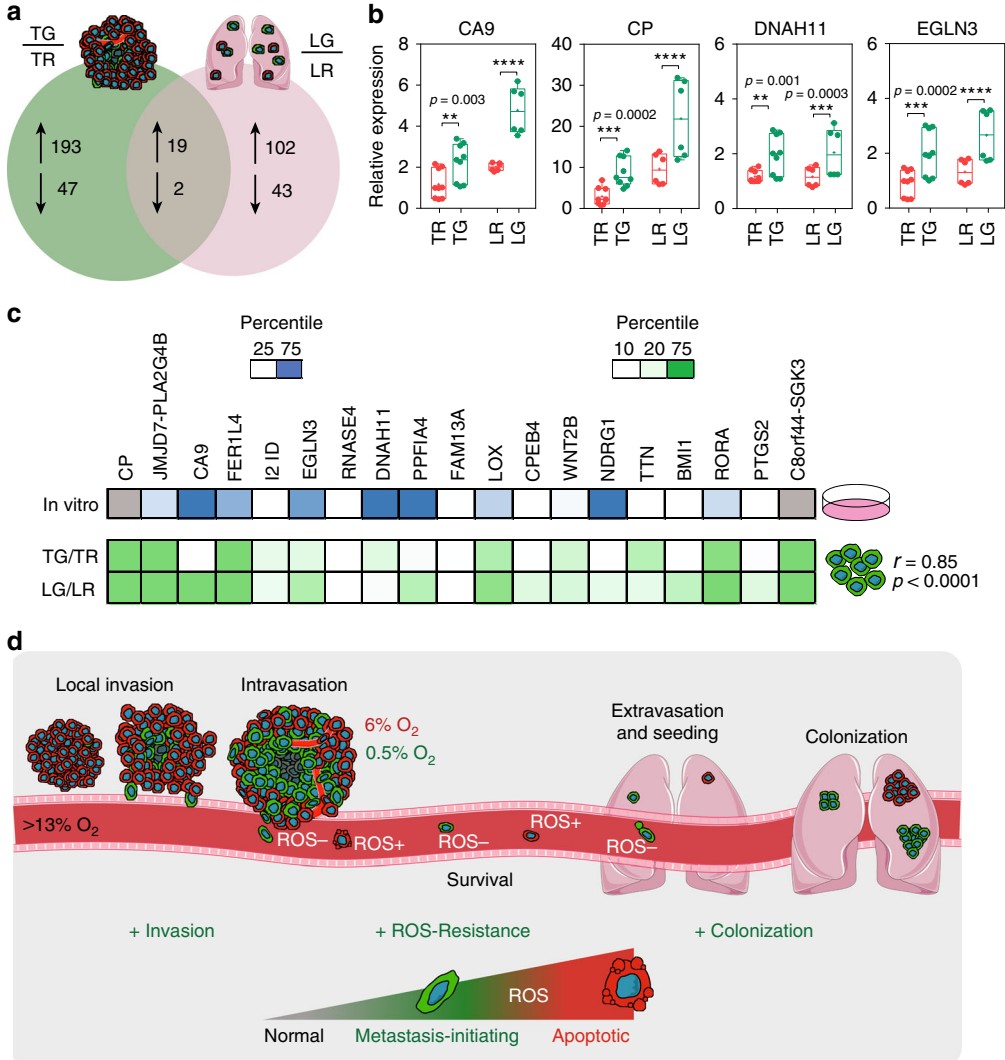

**Fig. 9** Post-hypoxic 'memory' at the metastatic site. **a** Venn diagram displaying the overlap of the number of genes with differential expression ($-1.5 \geq$ FC $\geq 1.5$) in GFP+ (TG) versus DsRed+ (TR) tumor cells (green circle) and GFP+ (LG) versus DsRed+ (LR) metastatic cells in the lung (pink circle) (Supplementary Table 9). **b** Relative mRNA expression was confirmed with independent samples by using qPCR in tumor cells (TR and TG) and metastatic cells in the lung (LR and LG) (mean ± SEM, $N = 2$–$3$, $n = 3$); ****$P < 0.0001$ TG versus TR and LG versus LR (two-tailed $t$-test). Primer sequences are available in Supplementary Table 1. The box extends from the 25th to 75th percentiles, the median is marked by the vertical line inside the box, and the whiskers represent the minimum and maximum points. **c** Heat map of the 19-gene signature derived from the overlap of tumor and lung GFP+ induction. The distribution of the relative fold change of each gene in the 19-gene signature is displayed for in vitro hypoxic exposure (in vitro), GFP+ versus DsRed+ sorted tumor cells (TG/TR), and GFP+ versus DsRed+ sorted lung metastatic cells (LG/LR). Genes with fold change higher than 75% of the fold change of genes in the in vitro set are blue and genes with fold change lower than 25% of the fold change of genes are white. Genes with fold change higher than 75% of the fold change of genes in the in vivo sets are dark green and genes with fold change lower than 10% of the fold change of genes are white (Pearson correlation factor TG/TR vs. LG/LR $r = 0.85$ and $P < 0.0001$). **d** Overview of the role of post-hypoxic cells in the metastatic cascade. Post-hypoxic cells (GFP+) have enhanced metastatic potential associated with enhanced invasion, metastatic initiating capacity, and a ROS-resistant phenotype that improves survival in the bloodstream

A global gene expression analysis demonstrated that the intratumoral exposure versus in vitro exposure to hypoxia results in distinctly different gene expression profiles. In vivo, oxygenation is finely tuned across time and length scales, reflecting important parameters such as the dilation and constriction of nearby blood vessels, changes in respiratory rate, and vascular remodeling. Hence, our result is not entirely surprising since nearly all of these processes, including interactions with the tumor microenvironment, are absent in cell culture. We found that the 41-gene signature derived from the overlap of intratumoral and in vitro hypoxia serves as a prognostic indicator of DMFS.

Although many hypoxia-inducible genes have been implicated in metastasis[63], prior studies have not considered whether the hypoxia-induced genes maintain their expression upon reoxygenation during the metastatic process such as upon exposure to $O_2$ in the bloodstream. Our in vitro exposure to hypoxia and reoxygenation shows that hypoxic gene expression is completely reversible with 48 h of reoxygenation, arguing that hypoxic gene regulation is short lived once the hypoxia stimulus is removed (Supplementary Fig. 8c). This is also consistent with the <30-min half-life of HIF-1α upon reoxygenation. However, nine of the nineteen RNA transcripts that are regulated in common in the GFP+ versus DsRed+ cells in the primary tumor and metastatic

lung lesions are hypoxia-regulated and HIF-inducible, suggesting a potential 'hypoxic memory'. This result suggests that a limited number of hypoxia-inducible genes maintain the same magnitude of increased expression at the metastatic site. Further work is warranted to determine the mechanistic underpinnings of this finding.

In summary, we described and characterized a hypoxia fate-mapping system that can be applied to address a wide range of biological questions in cancer research, embryonic development, as well as in pathologies that are critically affected by hypoxia. Moreover, we fully characterized the fate of hypoxic cells during cancer progression to metastasis, demonstrating that post-hypoxic cells have enhanced metastatic potential associated with enhanced invasion, metastatic initiating capacity, and a ROS-resistant phenotype (Fig. 9d). Future work is needed to determine whether post-hypoxic cells at the metastatic site are more resistant to current chemotherapy regimens, and whether targeting post-hypoxic cells could have a therapeutic benefit for patients with metastatic disease.

## Methods

**Cell culture**. Mycoplasma-free breast cancer cell lines, MDA-MB-231 (ATCC® HTB-26™), MCF7 (ATCC® HTB-22™), and 4T1 (ATCC® CRL-2539™) cells were obtained from the American Type Culture Collection (ATCC) and maintained in DMEM (Sigma-Aldrich) or RPMI-1640 (4T1; Sigma-Aldrich) with 10% FBS (Corning) and 1% penicillin/streptomycin (Invitrogen). Hypoxic conditions were achieved by using an InvivO$_2$ workstation (Baker) and an ICONIC (Baker) electronically controlled gas-mixing system maintained at 37 °C and 75% humidity, equilibrated at 1 or 0.5% O$_2$, 5% CO$_2$, and 94 or 94.5% N$_2$.

**Hypoxia fate-mapping construct design**. The loxp-DsRed-loxp-eGFP sequence was PCR amplified from the plasmid pMSCV-loxp-DsRed-loxp-eGFP-Puro-WPRE (#32702, Addgene) by using primers containing Sal1 and Not1 restriction enzyme sites. The amplified PCR fragment was digested with Sal1 and Not1, treated with Calf intestinal phosphatase (New England BioLabs), and agarose-gel purified. The fragmented DNA was ligated into the pENTRA1 vector cut with Sal1 and Not1. The Gateway System™ (Invitrogen) was used to recombine the pENTR1A shuttle vector with pLenti CMV/TO Zeo DEST (644-1) (#17294, Addgene) generating lentiviral vector 1.

The sequence GTGTACGTG (1HRE) spaced with random 5 base pairs of nucleotides was used to generate tandem copies from 1 to 9 HREs that were directly synthesized by Integrated DNA Technologies (IDT) as gBlocks (Coralville). The TK minimal promoter followed by 770-bps β-globin intron sequence as well as a CRE and CRE-ODD nucleotide sequence were also independently synthesized as gBlocks (IDT). The pRRLSIN.cPPT.PGK-GFP.WPRE (#12252, Addgene) was digested with ECORV and BAMH1 to remove the PGK promoter and EGFP cassette. In-Fusion cloning (Clonetech) was used to ligate the HRE gblock (s), the β-globin gblock, and the CRE or CRE-ODD gblock into the linearized pRRLSIN.cPPT.PGK-GFP.WPRE vector to generate the vectors shown in Supplementary Fig. 1a.

**Lentiviral transduction and selection**. Lentiviral vector 1 encoding CMV-loxp-DsRed-loxp-eGFP or lentiviral vector 2 encoding 4xHRE-MinTK-CRE-ODD were co-transfected with plasmid psPAX2 (#12260, Addgene) and plasmid pMD2.G (#12259, Addgene) into 293T cells by using Polyjet (SL10088, SignaGen (r)). Filtered viral supernatant generated from lentiviral vector 1 was collected 48 h post transfection and added to MDA-MB-231, MCF7, or 4T1 cells with 8 μg/mL polybrene (Sigma-Aldrich) overnight. After 24 h in fresh media, zeocin (Invitrogen) was added to the medium of cells for selection (100–250 μg/mL). Following selection, cells were transduced with lentivirus encoding 4xHRE-MinTK-CRE-ODD. The cell lines were exposed to 5% O$_2$ and flow sorted to remove GFP+ cells. The flow-sorted cells were single-cell cloned and screened by image analysis and flow cytometry to identify clones that switched from DsRed expressing to GFP expressing under 0.5% but not at 3% O$_2$ conditions.

**Live-cell imaging**. Cells (1 × 10$^5$) were plated in a 6-well plate and incubated overnight under standard cell culture conditions. The following day, cells were incubated in a Lionheart FX Automated Microscope (BioTek Instruments) maintained at 37 °C and controlled humidity with an on-stage environmental control. The microscope was maintained inside a customized sealed chamber (McCoy) equilibrated to 0.5% O$_2$, 5% CO$_2$, and 94.5% N$_2$. Images were taken at 10× with an Olympus (UPLFLN 10XPh) phase objective every 2 h for 6 days in phase-contrast, RFP, and GFP channels. The RFP and GFP image sequences were then overlapped into one, and the complete sequence was converted into an .avi file (Supplementary Movie 1).

**Immunoblot assays**. Aliquots of whole-cell lysates were prepared in NP-40 buffer (150 mM NaCl, 1% NP-40, and 50 mM Tris-HCl, pH 8.0) and fractionated by 10% sodium dodecyl sulfate–polyacrylamide gel electrophoresis (SDS–PAGE). Proteins were transferred from the SDS gel to a nitrocellulose membrane for 10 min by using a Trans-blot Turbo (Bio-Rad). The nitrocellulose membrane was blocked in 5% milk (% w/v) in Tris-buffered saline and 0.1% Tween-20 (TBS-T) for 30 min and incubated overnight with primary antibodies (1:1000) against HIF-1α (#610959, BD Biosciences) and GFP (#ab13970, Abcam). After washing three times with TBS-T, the nitrocellulose membrane was incubated with the corresponding anti-mouse (1:2500) (#AC2115, Azure Biosystems) or anti-chicken (1:5000) (AP162P, Sigma-Aldrich) HRP-secondary antibody for 1.5 h at room temperature with orbital shaking. β-actin was detected with a β-actin HRP-conjugated antibody (1:10,000) (#HRP-60008, Proteintech). ECL (Enhanced Chemiluminescent Substrate) (Perkin Elmer) was utilized as the substrate for HRP-catalyzed detection. The chemiluminescent signal was imaged by using a c300 imager (Azure Biosystems). DsRed-only and GFP-only cell lines were used as control for the GFP antibody.

**Hypoxia fate-mapping transgenic mouse**. The 4xHRE-MinTK-CRE-ODD sequence was cut from the previously described lentiviral vector 2 (Fig. 1b) and inserted via pronuclear injection into mouse embryos (Cyagen). We obtained 6 founder lines that we then bred to a tdTomato-floxed GFP (mT/mG) mouse (Jackson Labs; 007676). Genomic DNA was generated from ear punches and lysed in lysis buffer (1 M Tris, pH 8.0, 5 M NaCl, 0.5 M EDTA, 10% Tween-20, 10% NP-40, and ~40 μg of proteinase K) and incubated overnight at 37 °C. Proteinase K was deactivated at 95 °C for 2 min. Female double-transgenic hypoxia fate-mapping mice were then bred to an FVB/N-Tg(MMTV-PyMT) (Jackson Labs; 002374) male mouse to generate triple-transgenic female. Genotyping primers can be found in Supplementary Table 2.

Mice were i.p. injected with 1.25 mg of pimonidazole in saline (12.5 mg/ml) (Hypoxyprobe™-1) 1 h prior to sacrificing. Mammary fat pads located on the left side of the mouse were formalin-fixed, paraffin-embedded, and H&E stained. MFPs located on the right side of the mouse were fixed in 4% PFA at 4 °C for 4 h, saturated in 30% sucrose (Sigma-Aldrich) at 4 °C overnight, embedded in OCT media (Fisher Scientific), flash frozen in liquid nitrogen, sectioned via a cryotome CM1100 (Leica), and mounted onto Superfrost Plus Microscope Slides (Fisher Scientific). To evaluate the GFP and tdTom distribution, tumor tissue was stained with DAPI (1:1000 for 15 min, RT) and mounted with anti-fade solution. Bright-field images of H&E-stained sections were taken with an Olympus (UPLFLN 10XPh) phase objective 10× (BioTek Instruments). GFP and tdTom expression were evaluated by using frozen tissue mounted on slides and imaged by using a 40×/1.30 PlanNeofluar oil objective with DIC0X/0.45 PlanApo (dry, no DIC) and a 63×/1.4 PlanApo oil objective with DIC in a Zeiss LSM780-FCS microscope.

**TLA sequencing**. Bone marrow cells were collected from a 4xHRE-MinTK-CRE-ODD transgenic mouse. Bone marrow cells were flushed with PBS from 2 femurs. The cells were washed with PBS and frozen in PBS supplemented with 10% FBS and 10% DMSO. Viable frozen bone marrow cells were submitted and processed by Cergentis following their TLA (Targeted Locus Amplification) protocol[32]. Two primer sets were designed based on the nucleotide sequence of the transgene (TG) (Supplementary Table 3). The primer sets were used in individual TLA amplifications. PCR products were purified and library prepped by using the Illumina Nextflex protocol, and next-generation sequencing (NGS) was performed on an Illumina MiniSeq®. Reads were mapped by using BWA-SW (version 0.7.15-r1140, settings bwasw -b 7)[64], and the NGS reads were aligned to the transgene (TG) sequence and host genome. The mouse 10 mm genome was used as host reference genome sequence. The presence of single-nucleotide variants was determined by using samtools mpileup (samtools version 1.3.1)[65,66]. Fusion sequences, consisting of two parts of the TG, were identified by using a proprietary Cergentis script. Integration sites were detected based on coverage peaks in the genome and the identification of fusion reads between the TG sequence and host genome.

**Spheroid culture**. Cell spheroids were formed in two steps—spheroid formation in round-bottom 96-well tissue culture plates followed by spheroid embedment in a collagen gel[67,68]. A mixture of 80–90% wild-type (non-labeled) cells and 10–20% hypoxia fate-mapping cells were plated per well in spheroid formation media (2 × 10$^4$)—DMEM (Sigma-Aldrich) and Methocult H4100 (STEMCELL Technologies). Plates were centrifuged at 300g for 7 min, rotated, and centrifuged again. Following 72 h of incubation, spheroids were embedded into 2 mg/ml collagen containing DMEM and soluble rat tail type I collagen (Corning). Briefly, each spheroid was transferred to a Petri dish, where it was individually isolated with a collagen solution mix and quickly transferred to the center of a semi-cross-linked collagen gel in a 96-well plate at 37 °C. After complete cross-link, warm media was added. Spheroids were imaged in an environmentally controlled microscope every 2 days by using an Olympus (UPLFLN 4 ×) objective in Cytation 5 (BioTek Instruments). Multiple images were captured in order to display the entire spheroid (up to a 3 × 3-tile size). Confocal microscopy was performed to obtain z projections of spheroids by using a 10 × /0.45 PlanApo (dry, no DIC) objective in a Zeiss LSM780-FCS microscope. Z stacks spaced at 6.3-μm intervals of 4 × 4 tiles were

processed into a 3D image via Imaris version 9.2 (Bitplane), and 3D surface rendering was used to visualize the color distribution in 3D.

For the ex vivo invasion assay, after 4 days in culture, spheroids were imaged in an environmentally controlled microscope every 5 min for 16 h by using an Olympus (UPLFLN 4×) objective in Cytation 5 (BioTek Instruments). Cell trajectories were tracked by using MetaMorph software to obtain x, y coordinates at each time. Trajectories were fit by using the anisotropic persistent random walk (APRW) model in MATLAB[54] to determine total diffusivity and persistent time. By using NIS-Elements software, the same circular region of interest (ROI) was aligned with the center of each spheroid, and cells outside the ROI were counted as cells at the invasive front of the spheroid.

**Organoid culture.** Mammary organoids were derived from transgenic mouse tumors following previously published protocols[69]. Briefly, tumors from transgenic mice were mechanically disrupted by using a blade, followed by enzymatic digestion with 2 mg/ml collagenase at 37 °C in an orbital shaker (Sigma-Aldrich) for 1 h. The suspension was centrifuged at 520 g for 10 min, and the supernatant was discarded. Organoids were then digested with DNAse (10 mg/ml) (Sigma-Aldrich) for 5 min at RT. The suspension was spun down and resuspended in fresh media, followed by a differential centrifugation (×4) at 520 g for 2 seconds. Organoids were either frozen or embedded in Matrigel (Corning) at a density of 100–200 organoids/ml and plated in a 24-well plate (100 μl/well). Organoids were cultured with FGF-supplemented (40 ng/ml) (Sigma-Aldrich) media and imaged over time using Cytation 5 with an Olympus (UPLFLN 10XPh) phase objective (BioTek Instruments). Image-iT™ Hypoxia Reagent (Thermo Fisher Scientific) was added to the media to a final concentration of 5 μM, and the organoids were incubated at 37 °C for 3 h. Organoids were imaged with an Olympus (UPLFLN 10XPh) phase objective 10 × (BioTek Instruments). Confocal microscopy was performed to obtain z projections of organoids by using a 20×/0.80 PlanApo (dry, no DIC) objective in a Zeiss LSM780-FCS microscope. Z stacks were processed via Imaris version 9.2 (Bitplan), and 3D surface rendering was conducted to help the visualization of the color distribution. For flow cytometry analysis, organoids' gels were incubated with trypsin for 10 min, washed twice with PBS, and collected in FACS buffer. GFP was detected in the FITC channel, and DsRed in the PE channel by using a SH800S Cell Sorter manufactured by Sony Biotechnology. Organoids derived from mT/mG and MMTV-PyMT transgenes (2 T) were used as tdTom+/GFP– control. Organoids derived from a 2 T mouse were treated in suspension with adeno-cre (ad-cre) and used a GFP+ control. As previously reported[70], the transduction with ad-cre is not 100% successful in organoids, but was sufficient to obtain a GFP+ /tdTom– population for gating purposes. For RNA extraction, organoids' gels were mechanically dissociated with Tris Reagent (Zymo). Samples were then processed by using Direct-zol RNA kit (Zymo) with DNase I treatment. Actin was used as internal control. Primer sequences are available in Supplementary Table 1.

**Animal studies.** Animal research complied with all relevant ethical regulations according to protocols approved by the Johns Hopkins University Animal Care and Use Committee.

Female 5- to 7-week-old NOD-SCID Gamma (NSG) mice were anesthetized by the intraperitoneal injection (i.p.) of 100 mg/kg Ketamine, 16 mg/kg Xylazine, Vet One. In total, $2 \times 10^6$ MDA-MB-231 or $1 \times 10^5$ 4T1 hypoxia fate-mapping cells were injected into the mammary fat pad (MFP) closest to the second nipple. Intraductal injections were performed following previously published protocols[71]. Briefly, mice were anesthetized with isoflurane; $5 \times 10^4$ MCF7 hypoxia fate-mapping cells were injected directly into the mammary duct of 8- to 12-month-old multiparous (NSG) female mice. Mice were i.p. injected with 1.25 mg of pimonidazole in saline (12.5 mg/ml) (Hypoxyprobe™-1 kit) 1 h prior to sacrificing.

Tumors were excised at various time points, formalin-fixed (Sigma-Aldrich) for 1 h and saturated in 30% sucrose (Sigma-Aldrich) at 4 °C overnight, and embedded in OCT media (Fisher Scientific), flash frozen in liquid nitrogen, sectioned via a cryotome CM1100 (Leica), and mounted onto Superfrost Plus Microscope Slides (Fisher Scientific). To evaluate the GFP and DsRed/tdTom distribution, tumor tissue was stained with DAPI (1:1000 for 15 min, RT) and mounted with anti-fade solution. To assess the entire cross section of the tumor, each slide was imaged with an Olympus (UPLFLN 4×) objective by using Cytation 5 microscope (BioTek Instruments). Multiple image tiles were linearly stitched with Gen5 Software (BioTek Instruments).

**Tumor removal surgery.** Tumor removal surgery was performed 12 days (4T1 model) or 2 weeks (MDA-MB-231 model) after tumor implantation. Mice were anesthetized and transferred to a heating pad. After hair removal, a small incision was made, and the tumor was carefully detached from adjacent skin, adjacent loose tissue and visible lymph nodes were also removed. The wound was closed by using 9-mm autoclips (Braintree Scientific, Inc.). Betadine was applied to the wound, and ophthalmic ointment was used to prevent eye desiccation.

**Analysis of metastatic progression.** An average of 500 μl of blood was collected by cardiac puncture into an EDTA tube by using a 26-G syringe needle. Red blood cells were lysed by using cold ACK Lysing Buffer (Quality Biological). Cells were

incubated with 5 ml of the lysis buffer on ice for 10 min, followed by cold centrifugation at 1700 g for 5 min. This step was repeated, and the remaining cells were washed and resuspended in FACS buffer.

To assess and quantify metastasis, lungs were inflated with an OCT:PBS solution and excised for both image and flow cytometry analysis. For image analysis, lungs were formalin-fixed, saturated in 30% sucrose, embedded in OCT media, flash frozen in liquid nitrogen, sectioned via a cryotome CM1100 (Leica), DAPI-stained, and mounted for imaging. Full lung-slide sections were imaged as described for the primary tumor. Image analysis was performed by using NIS-Elements software. Individual metastatic events were carefully annotated. To measure metastasis size, metastatic events were marked as ROIs by using the auto-detect tool available in NIS-Elements. For flow cytometry analysis, lungs were chopped and digested (collagenase 2 mg/mL (Sigma-Aldrich), BSA 2 mg/mL (Gemini Bioproducts)) for 1 h at 37 °C at 160 RPM. After passing through a 70-μm strainer, cells were washed with PBS and resuspended in FACS buffer.

**Intravenous injection.** Animals were warmed for 5–10 min with an overhead heat lamp to dilate the veins. After applying friction to the tail, animals were restrained, $1 \times 10^5$ fate-mapping cells were injected, and mice were sacrificed at the time points indicated.

**Tumor and lung dissociation.** Tumors and lungs were harvested, digested, and processed as described above. The cell suspension was filtered (70-μm strainer), washed, and enriched by magnetic-activated cell separation (MACS) by using a Mouse Cell Depletion Kit (Miltenyi Biotec) following the manufacturer's instructions. The resultant cell suspension was then washed with PBS and suspended in sorting buffer. Cells were sorted by using a SH800 cell sorter (Sony Biotechnology) directly into Tris Reagent (Zymo Research) for RNA extraction followed by RNA sequencing or directly into cell culture media to establish cell lines.

**Fluorescence-activated cell sorting (FACS).** Cells were trypsinized, resuspended in culture media, fixed with 4% PFA (Fisher Scientific) for 5 min, washed with PBS, and collected in FACS buffer (PBS, 1% BSA, 0.5 mM EDTA, and 25 μg/ml DNAse). Samples were analyzed by using a FACSCalibur (BD Biosciences) flow cytometer. GFP was detected in the FL-1 channel, and DsRed was detected in the FL-2 channel. DsRed-only and GFP-only cell lines were used as controls for compensation. Data were analyzed via FlowJo V10 software (Tree Star, Inc.).

For cell sorting, cells were resuspended in sorting buffer (PBS, 1% BSA, 0.5 mM EDTA, and 25 mM HEPES, pH 8). Samples were sorted by using a FACSAria II (BD Biosciences) cell sorter cytometer into (GFP+/DsRed−) or DsRed+ expressing populations directly into media. GFP was detected in the FITC channel, and DsRed was detected in the PE channel. DsRed and GFP-only cell lines were used as controls for compensation by using FACSDiva (BD Biosciences) (Supplementary Fig. 1b).

**Oxygen measurements.** $O_2$ measurements were conducted by using a RED-FLASH system (FireStingO_2, Pyroscience). In spheroids with compact cores, a retractable-needle microsensor (50–70 μm) was used to measure $O_2$ level in the core and at the edge of the spheroid. $O_2$ levels in Matrigel (organoids) were performed by using the OXNANO oxygen nanoprobes dispersed in the Matrigel. $O_2$ levels were measured every 2 days with a REDFLASH oxygen meter from the bottom of the well. In vivo $O_2$ measurements were conducted on mouse tumors by using a fixed-needle (0.9 mm/230 μm) mini-sensor mounted on a manual micrometer. After determining the lowest $O_2$ level in the tumor core, the probe was slowly retracted, and $O_2$ measurements were recorded at 0.5- mm intervals until reaching the tumor edge (Supplementary Fig. 5c). Needle probes were carefully washed and calibrated to atmospheric $O_2$ before each measurement.

**Immunohistochemistry staining (IHC).** Paraffin-embedded tissue sections of an MDA-MB-231 tumor were dewaxed with Clarification new-clean® (EMD) and hydrated with a series of decreasing ethanol baths (100%, 80%, 80, and 70%) followed by water. Citrate-EDTA buffer was used for antigen retrieval at 85 °C for 40 min. After cooling to room temperature, LSAB+ System (Dako) was used for Hypoxyprobe™-1 staining according to the manufacturer's instructions and counterstained with hematoxylin (Sigma-Aldrich). Tissue was dehydrated and mounted.

**Immunofluorescence staining.** Immunofluorescence staining was performed in tissue cryo-sections after permeabilization with 0.1% Triton-X (Sigma-Aldrich) for 10 min for the detection HIF-1α. All slides were blocked with 2% BSA for 30 min. Slides were incubated overnight at 4 °C with primary antibodies against HIF-1α (#SC-10790, Santa Cruz Biotechnology) and/or Hypoxyprobe™-1 (dilutions at 1:50), or CD31 (#ab28364, Abcam) (dilution at 1:100). The slides were then incubated with the corresponding mouse (#A21237) or rabbit (#A21245) secondary Alexa Fluor 647™ (Invitrogen) antibodies at 1:1000 dilution for 1.5 h at RT, followed by DAPI staining (1:1000 for 15 min, RT). In order to obtain two immunofluorescence stains in the same tissue slide, a secondary Alexa Fluor 350™ (#A11045, Invitrogen) was used, and DAPI staining was not performed. Apoptotic cells were stained with Tunel labeling Kit (Thermo Fisher Scientific) according to

the manufacturer's instructions. All slides were mounted with anti-fade solution (90% glycerol, 20 mM Tris, pH 8.0, and 0.5% N-prolyl gallate). Slides were imaged by using a Cytation 5 (BioTek Instruments) or by using a LSM780-FCS laser-scanning confocal microscope (Zeiss) with a 40×/1.30 PlanNeofluar oil objective with DIC0X/0.45 PlanApo (dry, no DIC).

**RNA library preparation.** Total RNA was extracted from cells by using TRIzol (Invitrogen) and purified by using Direct-zol RNA mini kit (Zymo Research) with DNase I treatment. After RNA purification, samples were confirmed to have a RIN value > 9.0 when measured on an Agilent Bioanalyzer. Libraries for RNA sequencing were prepared with KAPA Stranded RNA-Sequencing Kit. Briefly, the workflow consisted of mRNA enrichment, cDNA synthesis, end repair to generate blunt ends, A-tailing, adapter ligation, and 14 cycles of PCR amplification. Unique adapters were used for each sample in order to multiplex samples into several lanes. Sequencing was performed on Illumina Hiseq 3000/4000 with a 150-bp pair-end run by Quick Biology (Pasadena, CA).

**RNA-sequencing analysis.** A data quality check was done on Illumina SAV. Demultiplexing was performed with Illumina Bcl2fastq2 v 2.17 program. The sequence reads were mapped to the latest UCSC transcript set by using Bowtie2 version 2.1.0[72], and the gene expression level was estimated by using RSEM v1.2.15[73]. TMM (Trimmed Mean of M-values) was used to normalize gene expression. Differentially expressed genes were identified by using the edgeR program[74]. Genes showing altered expression with $p < 0.05$ and more than 1.5-fold changes were considered differentially expressed (Supplementary Tables 4 and 7). Normalized (TMM) and raw data were uploaded to GEO.

Gene Set Enrichment Analysis (GSEA)[40] was conducted to determine the specific 'Hallmark' gene sets from The Molecular Signatures Database (MSigDB) collection[75] that showed statistically significant and concordant differences between GFP and DsRed (Supplementary Tables 5 and 8) by using the GSEA desktop application. TMM-normalized data were used as the expression dataset, the 'Hallmark' gene sets were selected as the gene sets database, and number of permutations used was set to the recommended (1000). Gene expression sets—1/20% O$_2$, TG/TR, and LG/LR—were compared by using the VennPlex tool[76] to determine the number of genes that overlap between the different conditions ($1.5 \geq$ FC $\geq 1.5$, $p$-value $\leq 0.05$) (Supplementary Tables 6 and 9). Pre-ranked GSEA analysis[40] was conducted to compare the level of enrichment in 'Hallmark' gene sets between cells that experience intratumoral hypoxia and those exposed to hypoxia in vitro. Matched gene lists were ranked based on fold change upon intratumoral or in vitro hypoxia, respectively (TG/TR or 1/20% O$_2$). The fold change of multiple samples was averaged before ranking the gene set. Default settings were used (enrichment statistic: classic; normalization mode: meandiv), and the number of permutations was set to 1000.

**Reverse transcription and quantitative real-time PCR.** Total RNA was extracted from cells or cells sorted from tumors by using Direct-zol miniRNA kit (Zymo) with DNase I treatment. Two-hundred-and-fifty to one-thousand nanograms of RNA was used for first-strand DNA synthesis with the iScript cDNA Synthesis system (Bio-Rad). qPCR was performed by using human-specific primers and iTaq SYBR Green Universal Master Mix (Bio-Rad). The expression of each target mRNA relative to 18S RNA was calculated based on the threshold cycle (Ct) as $2 - \Delta(\Delta Ct)$, where $\Delta Ct = Ct_{target} - Ct_{18S}$ and $\Delta(\Delta Ct) = \Delta Ct_{test} - \Delta Ct_{control}$. Each biological sample consisted of 3 technical repeats and 3 biological repeats were combined. Primer sequences are available in Supplementary Table 1.

**MitoROS and viability measurements.** Mitochondrial ROS levels and viability was measured by using Elite Mitochondrial ROS Activity Kit—Deep Red Fluorescence (MitoROS) (e-Enzyme) and Sytox Blue (Invitrogen) according to the manufacturer's instructions. Fluorescence intensity was measured by using a SH800 cytometer (Sony).

**Treatment with H$_2$O$_2$.** Tumor-derived cells sorted for DsRed+ or GFP+ were plated in 96-well plates (5000 cells/well). On the following day, fresh media with hydrogen peroxide (H$_2$O$_2$) (Sigma) was added. One hour later, cells were washed with PBS, and fresh media was added to the wells. Following a 48-h recovery period, cells were washed with PBS, fixed with 4% PFA, and stained with DAPI. Whole wells were imaged with an Olympus (UPLFLN 4XPh) phase objective 4× in a 4 × 3 montage (BioTek Instruments) in the RFP, GFP, and DAPI channels. Images were linearly stitched, and DAPI area was quantified by thresholding to determine well confluency using Gen5 (BioTek Instruments).

**Suspension assay.** Tumor-derived cells ($1 \times 10^5$) were resuspended in 1 ml of fresh media in a 1.5-ml centrifuge tube. Tubes were incubated at 37 °C, 5% CO$_2$ in a rotisserie rotator. Following 72 h, cells were plated in 6-well plates. On the following day, cells were washed, fixed with 4% PFA, and stained with DAPI. Confluency was assessed as described for the H$_2$O$_2$ treatment.

**Shear-stress assay.** This assay was adapted from published work[77]. Tumor-derived cells were resuspended in a cell suspension ($1 \times 10^5$/ml) that was slowly collected into a syringe without a needle attached. The suspension was then expelled through a 26-G needle at a constant flow rate of 0.25 ml/s. After 10 cycles, cells were counted by using Trypan Blue to asses viability.

**Statistical analysis.** All data are presented as mean ± SEM, and statistical analysis was performed by using GraphPad Prism 6 and statistical tests appropriate for each experimental setup. All comparisons between DsRed+ and GFP+ populations from the same tumor or organoid were tested by using a matched Two-way ANOVA with Bonferroni multi-comparison tests. A comparison of two variables was performed by using an unmatched Two-way ANOVA with Bonferroni multi-comparison tests. The comparison of a single variable across multiple experimental conditions was performed by using a one-way ANOVA with Bonferroni multi-comparison test. A comparison of a single variable measured in a sample at two different locations was performed via paired two-tailed Student's $t$-test. Significance levels are reported as ****$P < 0.0001$, ***$P < 0.001$, **$P < 0.01$, *$P < 0.1$.

**Reporting summary.** Further information on research design is available in the Nature Research Reporting Summary linked to this article.

## Data availability

The RNA-sequencing data have been deposited in the GEO database under the accession codes GSE111653 (in vitro hypoxia), GSE126609 (tumor), and GSE136372 (lung). The survival data referenced during the study are available in a public repository from the kmplot.com website. All the other data supporting the findings of this study are available within the article and its supplementary information files and from the corresponding author upon reasonable request. A reporting summary for this article is available as a Supplementary Information file.

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

## Acknowledgements

We thank Dr. Denis Wirtz, Dr. Pei-Hsun Wu, Dr. Andrew Ewald, Dr. Ben Ho Park, Dr. Saraswati Sukumar, Dr. Dipali Sharma, Soumitra Bhoyar, and Josh DiGiacomo for their feedback over the course of this study. We particularly thank Dr. Sukumar for guidance with the intraductal mouse model. Dr. Hanhvy Bui (IIC Flow Cytometry Core) and Dr. Jessica Gucwa (SKCCC Flow Cytometry Core) assisted with flow cytometry. Marc Rosen, Jisu Shin, and Shyanne Salen assisted with tail-vein injections, migration tracking and analysis, and organoid generation, respectively. Work in the Gilkes lab is supported by U54-CA210173 (NCI), R00-CA181352 (NCI), The V Scholar Foundation, Susan G. Komen Foundation (CCR17483484), The Jayne Koskinas Ted Giovanis Foundation for Health and Policy, Cindy Rosencrans Fund for Metastatic Triple-Negative Breast Cancer, The Emerson Collective, and the SKCCC Core Grant (P50CA006973 (NCI)). Cartoons included in the display items of this work were adapted from Servier Medical Art licensed under a Creative Commons Attribution 3.0 Unported License (smart.servier.com).

## Author contributions

I.G. was responsible for research design, execution, data analysis and interpretation, and paper preparation; Y.S., I.G. and I.Y. were responsible for the triple-transgenic mice screening; J.A.J. assisted on animal work and immune blotting; I.Y. and Y.S. were responsible for the transgenic mice genotyping; I.Y. assisted on the preparation of the RNAseq samples; G.W. performed the intraductal injections; D.M.G. was responsible for concept and research design, supervision, colony-breeding coordination, data interpretation, and paper preparation.

## Competing interests

The authors declare no competing interests.

## Additional information

**Peer Review Information** *Nature Communications* thanks Nicola Valeri and other, anonymous, reviewer(s) for their contribution to the peer review of this work. Peer reviewer reports are available.

