## [Peer Review File · Nature Communications]

REVIEWERS' COMMENTS:

Reviewer #1 (Remarks to the Author):

The authors have done a credible job in revising the paper. I also think it is a better fit for Nature Communications than Nature Cell Biology.

Reviewer #3 (Remarks to the Author):

The revised manuscript has significantly improved with the clarification of previous reviewers' points and the interesting new data on hypoxic cells in metastasis. Overall, this work should be of interest to the readers of Nature Comms.

The authors showed that a major effect of hypoxia on metastasis is to promote the intravasation of tumor cells into the blood stream. This is at least in part mediated by increased invasiveness as shown in Fig. 7. The other major new finding of the manuscript is that in vivo hypoxic cells have different gene expression profiles, presumably properties, from in vitro hypoxic cells. Therefore, it will be interesting to test whether hypoxic cells generated in vitro culture are not as invasive as hypoxic cell in vivo in Fig. 7.

Reviewer #4 (Remarks to the Author):

All the comments from the previous round of review at Nat Cell Biology have been addressed

Nicola Valeri

REVIEWERS' COMMENTS:

Reviewer #1 (Remarks to the Author):

The authors have done a credible job in revising the paper. I also think it is a better fit for Nature Communications than Nature Cell Biology.

We thank the reviewer for taking the time to review our work and provide valuable feedback.

Reviewer #3 (Remarks to the Author):

The revised manuscript has significantly improved with the clarification of previous reviewers' points and the interesting new data on hypoxic cells in metastasis. Overall, this work should be of interest to the readers of Nature Comms.

The authors showed that a major effect of hypoxia on metastasis is to promote the intravasation of tumor cells into the blood stream. This is at least in part mediated by increased invasiveness as shown in Fig. 7. The other major new finding of the manuscript is that in vivo hypoxic cells have different gene expression profiles, presumably properties, from in vitro hypoxic cells. Therefore, it will be interesting to test whether hypoxic cells generated in vitro culture are not as invasive as hypoxic cell in vivo in Fig. 7.

In our previous work (Ju *et al.*, 2019), we reported that hypoxia increases invasion in spheroids. Spheroids composed of MDA-MB-231 cells were fully embedded into 2 mg/mL collagen gels and under 20% or 1% O₂ for 5 days. At the endpoint of the experiment, we found that the invaded distance was significantly higher in cells cultured under hypoxic spheroids. We have not tested cells that have been exposed to hypoxia in vitro and then re-oxygenated.

We thank the reviewer for the interesting suggestions and feedback.

Reviewer #4 (Remarks to the Author):

All the comments from the previous round of review at Nat Cell Biology have been addressed

Nicola Valeri

We thank Dr. Nicola Valeri for taking the time to review our work and provide valuable feedback.